# KRAS regulation by small non-coding RNAs and SNARE proteins

Yonglu Che[1,2], Zurab Siprashvili[1,2], Joanna R. Kovalski[1,2], Tiffany Jiang[1], Glenn Wozniak[1], Lara Elcavage [1] & Paul A. Khavari [1,2,3]*

KRAS receives and relays signals at the plasma membrane (PM) where it transmits extracellular growth factor signals to downstream effectors. SNORD50A/B were recently found to bind KRAS and inhibit its tumorigenic action by unknown mechanisms. KRAS proximity protein labeling was therefore undertaken in *SNORD50A/B* wild-type and knockout cells, revealing that SNORD50A/B RNAs shape the composition of proteins proximal to KRAS, notably by inhibiting KRAS proximity to the SNARE vesicular transport proteins SNAP23, SNAP29, and VAMP3. To remain enriched on the PM, KRAS undergoes cycles of endocytosis, solubilization, and vesicular transport to the PM. Here we report that SNAREs are essential for the final step of this process, with KRAS localization to the PM facilitated by SNAREs but antagonized by SNORD50A/B. Antagonism between SNORD50A/B RNAs and specific SNARE proteins thus controls KRAS localization, signaling, and tumorigenesis, and disrupting SNARE-enabled KRAS function represents a potential therapeutic opportunity in KRAS-driven cancer.

[1] Program in Epithelial Biology, Stanford University, Stanford, CA 94305, USA. [2] Program in Cancer Biology, Stanford University, Stanford, CA 94305, USA. [3] VA Palo Alto Healthcare System, Palo Alto, CA 94304, USA. *email: khavari@stanford.edu

Ras superfamily GTPases are enzymatic switches that serve as signal relays in a variety of essential eukaryotic pathways[1]. These GTPases exist in either GTP or GDP-bound states with an intrinsic mechanism to hydrolyze GTP to GDP. Mutations in human Ras genes that inhibit hydrolysis keep the protein GTP-bound and thus competent to constitutively activate downstream signaling pathways, including those mediated by Raf and PI3K family kinases[2–4]. Altogether, the three human Ras genes, KRAS, NRAS, and HRAS, are among the most frequently mutated oncogenes in cancer, cumulatively estimated at 25–30% of all malignancies. KRAS accounts for 85% of these Ras isoform mutations. Among cancers responsible for the most cancer-associated deaths in the United States are three that are primarily driven by oncogenic KRAS mutations; pancreatic ductal adenocarcinoma (95% with oncogenic KRAS mutations), colorectal adenocarcinomas (52%), and lung adenocarcinomas (31%)[5–8]. Experimental evidence and analysis of human syndromes caused by germline Ras mutations supports the observation that KRAS is the strongest oncogene in the family. The KRAS oncoprotein, but not HRAS or NRAS, confers stem-like properties on cell lines and fully activated KRAS alleles are not tolerated in development whereas activated HRAS alleles manifest in cardiofacial-cutaneous syndromes. Strategies to inhibit Ras have included efforts to directly inhibit the protein, disrupt its membrane localization, target downstream effector pathways, exploit synthetic lethal interactions, and perturb Ras-regulated metabolic processes[9]. Thus far, despite over 30 years of investigation, there is no clinically effective anti-Ras therapy.

To relay extracellular growth factor binding from receptor tyrosine kinases to effector pathways that propagate pro-tumorigenic signaling, Ras must localize to a subcellular space where it can both interface with plasma membrane (PM)-bound receptors and recruit complexes necessary for downstream signal activation[10]. To achieve proper localization, all Ras proteins are covalently modified with a C-terminal lipid farnesyl group that increases membrane affinity and a methylation modification at the same site to reduce charge-based membrane repulsion[3]. The solubilizing prenyl-binding protein phosphodiesterase δ (PDEδ) then facilitates efficient deposition of Ras onto endomembrane spaces, and Ras is then shuttled to the plasma membrane through vesicular transport. A key structural difference between KRAS and its sister isoforms is embedded in the nuances of subcellular transport. While all Ras proteins undergo a farnesyl modification on the most C-terminal cysteine residue, HRAS and NRAS have an additional site that can be acylated to improve membrane association[11]. In contrast, the same site on KRAS cannot be modified but instead contains a stretch of positively-charged lysines, termed the polybasic region, that mimic permanent acylation. It is likely that differences arising from removable acylation versus those mimicking permanent acylation force KRAS to take an alternate route to the PM. Understanding the mechanisms of Ras isoform trafficking within the cell will thus fill a major present gap in knowledge about Ras biology as well as potentially identify alternative treatment strategies.

Vesicular transport is a carefully orchestrated cellular process that is responsible for compartment integrity, exocytosis, and trafficking within the cell. The SNARE (Soluble NSF Attachment Protein Receptor) protein superfamily includes 38 protein members in humans that reside on membrane surfaces to direct and target fusion of vesicles with their proper target membrane[12]. Canonically, SNAREs initiate an energetically demanding zippering process where complementary SNARE proteins drive vesicle-target membrane fusion. This fusion event results in release of vesicle-contained cargo into the space beyond the target membrane as well as inclusion of surface-bound proteins into the target membrane[13]. SNAREs are best studied in the context of synaptic vesicle fusion in neurons where release of neurotransmitters from vesicles into the synaptic gap is crucial for intercellular signaling. Neurotransmitter release is notably inhibited by classes of botulism and tetanus toxins that cleave SNARE proteins[14]. These toxin proteases effectively shut down synaptic vesicle transport by direct degradation of the SNARE protein machinery critical for membrane fusion. While vesicular transport has been implicated in the trafficking of Ras isoforms to their sites of active signaling[15], the specific transporter proteins involved are unknown.

Recent work characterizing small non-coding RNAs in cancer identified an unexpected role for specific snoRNAs in the control of KRAS-driven tumorigenesis. Analysis of 5473 tumor-normal genome pairs revealed deletion of a pair of highly homologous ~70 nucleotide small nucleolar RNAs (snoRNAs), SNORD50A/B, in 10–40% of 12 human cancers, as well as shortened survival in patients with SNORD50A/B deletion[16]. Previous studies had identified SNORD50A/B at a chromosomal breakpoint in B-cell lymphoma and a germline 2-bp deletion in SNORD50A that conveys increased risk for prostate cancer[17,18]. Hybridization of SNORD50A/B snoRNAs to microarrays containing ~9200 recombinant proteins returned KRAS among the strongest binders, a result confirmed by electromobility shift assays and crosslinking and immunoprecipitation (CLIP). SnoRNAs are classically thought to reside within the nucleolus and recruit protein complexes to direct ribosomal RNA modification[19]; however, KRAS association with SNORD50A/B occurred outside the nuclear compartment, consistent with recent work demonstrating non-canonical snoRNA functions in an array of diseases, including Prader–Willi syndrome and cancer[20–26]. SNORD50A/B was found to act outside of classically defined snoRNA functions by suppressing KRAS[16]. In this setting, SNORD50A/B deletion resulted in hyperactive ERK signaling as well as accelerated tumorigenesis in vivo. How SNORD50A/B RNAs regulate KRAS, however, has been unclear.

Here, we use proximity proteomics in SNORD50A/B wild-type and knockout cells to uncover an essential role for SNAP23, SNAP29, and VAMP3 SNARE vesicular transport proteins in KRAS delivery to the PM and in KRAS-driven signaling, gene regulation, and tumorigenesis. These specific SNARE proteins, which have not previously been associated with KRAS, compete with SNORD50A/B snoRNAs for KRAS binding to at the same KRAS amino acid residues and exert antagonistic effects on KRAS trafficking and function. In this context, SNAP23, SNAP29, and VAMP3 are required for KRAS re-localization from the recycling endosome to the PM, and for KRAS association with its upstream receptors and downstream kinases, whereas SNORD50A/B snoRNAs oppose this process. In vivo tumorigenesis experiments with 15 cancer cell lines in a variety of genetic backgrounds revealed striking sensitivity of tumors driven by oncogenic KRAS, but not other Ras isoforms, to loss of these vesicular transport proteins. Moreover, analysis of human TCGA data demonstrated that loss of these specific SNARES, but not other SNARE family members, was associated with improved clinical survival in KRAS mutant pancreatic cancer, suggesting that oncogenic KRAS depends upon intactness of these SNAREs to achieve its full malignant impacts. Finally, a catalytic light chain of botulism toxin E protease engineered to cleave SNAP23 suppresses in vivo tumorigenesis by KRAS-driven tumor cells, indicating that vesicular transporters may provide a therapeutic opportunity in KRAS-dependent malignancies.

## Results

**SNORD50A/B deletion enhances KRAS proximity to specific SNAREs.** To begin to investigate the basis for the inhibitory

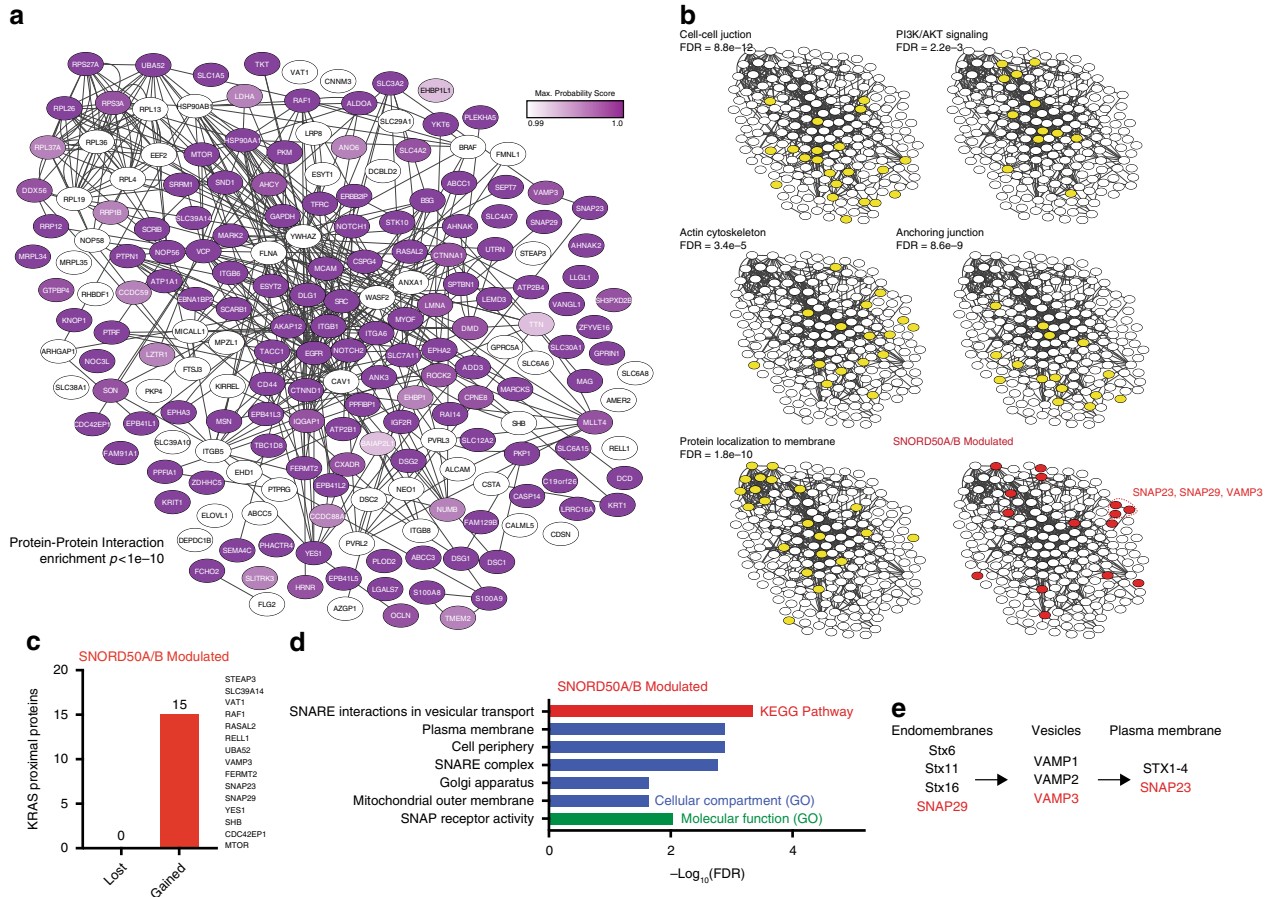

**Fig. 1** SNORD50A/B shapes the proximal proteome of KRAS. **a** Top 197 KRAS proximal proteins identified by mass spectrometry plotted using Stringdb. Edges represent either known protein-protein interactions or pathway interactions (3 SNORD50A/B WT cancer cell lines [H23, A549, and CHL1], 6 SNORD50A/B KO clones). **b** Duplicate interaction networks from **a** highlighting the indicated annotations identified by Genemania (yellow) or by our analysis (red). **c** KRAS proximal proteins that are modulated by SNORD50A/B resolved by whether they are lost or gained upon SNORD50A/B knockout (SAINT probability score change of >0.25, replicable >1.5 fold change). **d** GO Terms and KEGG pathway interactions associated with the SNORD50A/B-modulated interactome. **e** Abbreviated subset of the KEGG pathway: SNARE interactions in vesicular transport. KRAS interactors identified in this study highlighted in red

effect of SNORD50A/B snoRNAs on KRAS, the effect of SNORD50A/B on the composition of the proteins proximal to KRAS was examined. To generate a broad landscape of the KRAS proximal interactome, duplicate SNORD50A/B-knockout (KO) subclones were generated from three independent human cancer cell lines (H23, A549, CHL1). The generation, validation, and quality control of these lines is described in detail in our previous work[16]. In each of the nine total cell lines (3 wild-type [WT], 6 SNORD50A/B KO), a fusion of the BirA* promiscuous biotin ligase to KRAS was expressed at physiologic levels in an N-terminal design that preserves Ras function and localization[27] (Supplementary Fig. 1A–E). Since the KRAS4B isoform is the more common variant associated with human cancer, the KRAS4B sequence was cloned into the BirA* fusion construct and unless otherwise noted, KRAS in this manuscript refers to KRAS4B. BirA* releases reactive biotin intermediates within a radius of 10–20 nm[28,29], thereby biotin-labeling proteins proximal to KRAS in these cells. Proteins biotinylated by BirA*-KRAS were identified by streptavidin pulldown and mass spectrometry (Supplementary Data 1). This revealed a highly-interconnected set of proteins enriched in annotations consistent with Ras biology, including membrane localization, the ribosome, and PI3K/AKT signaling[30] (Fig. 1a, b; Supplementary Fig. 2A–F).

15 proteins differed significantly between SNORD50A/B WT and KO lines, as assessed by SAINT probability scores and replicability in multiple cell lines (Fig. 1c), suggesting that SNORD50A/B snoRNAs modulate KRAS protein associations. Among these candidate SNORD50A/B-modulated interactors, all 15 showed increased KRAS proximity in SNORD50A/B KO cells compared to WT suggesting that SNORD50A/B binding to KRAS may inhibit KRAS proximity to these proteins. These 15 proteins were enriched in functional annotations for a single significant KEGG pathway: SNARE interactions in vesicular transport (Fig. 1d). To verify that these changing interactions were not simply secondary to the known expression changes induced by the role of the snoRNA on mRNA processing, the KRAS-proximal interactors were compared to a prior study cataloging the transcriptional changes downstream of SNORD50A KD, revealing no overlap in the two sets. This suggests that these interaction changes are not explained by SNORD50A/B's role in mRNA processing (Supplementary Fig. 2H). The specific proteins annotated in this pathway and simultaneously enriched in the KRAS-proximal proteome were SNAP29, VAMP3, and SNAP23 (Fig. 1e), a series of SNARE proteins implicated in endosome-to-plasma membrane (PM) traffic. The proximal interactions of these SNAREs were assessed by a similar experimental design for KRAS, HRAS, and NRAS in cells derived from the most cancer-

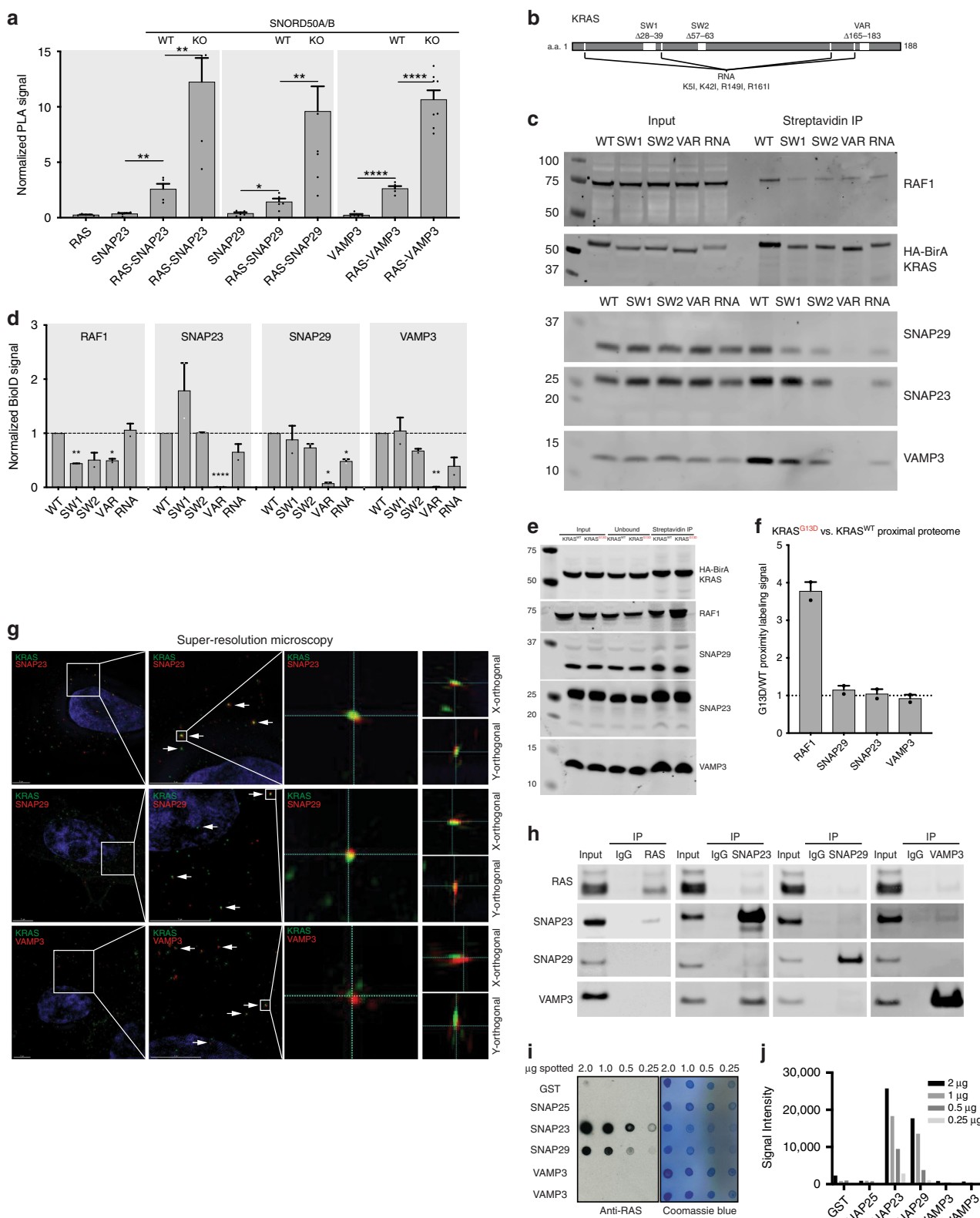

relevant tissues for each isoform. This revealed a KRAS-preferential proximity of SNAP23 (Supplementary Fig. 2G). SNARE protein proximity to KRAS in cells was confirmed via proximity ligation assay (PLA)[31,32]. SNAP23, SNAP29, and VAMP3 all co-localize with KRAS at a basal level, even in SNORD50A/B wild-type cells, but this co-localization greatly increases in the absence of these RNAs (Fig. 2a). SNORD50A/B

RNAs thus modulate KRAS proximity to specific SNARE transport proteins.

The regions of the KRAS protein required for SNARE protein adjacency were next examined. Proximity labeling was repeated with mutations or deletions in KRAS functional domains. Deletions of either of the KRAS switch I or switch II regions involved in effector binding reduced KRAS association with Raf1

**Fig. 2** SNARE interactions with KRAS are selectively SNORD50A/B-dependent. **a** Proximity ligation analysis (PLA) between KRAS and either SNAP23, SNAP29, or VAMP3, as well as single antibody controls in H23. Significance of WT cells determined by comparison to single antibody controls while significance of KO cells determined by comparison to WT. **b** Schematic of KRAS mutants used in proximity protein labeling experiments. **c** Strength of interaction between Raf1, SNAP23, SNAP29, or VAMP3 with KRAS mutants measured by proximity protein labeling followed by pulldown western blotting ($n = 2$) in H23. **d** Quantitation of **c**. **e** Interaction strength of Raf1 and SNAREs with KRAS$^{WT}$ and KRAS$^{G13D}$ in H23. **f** Quantification of **e** ($n = 2$). **g** Localization of HA-tagged KRAS and endogenous SNAP23, SNAP29, and VAMP3 visualized by super-resolution microscopy in A549. Scale bars are 5 µm. **h** Co-immunoprecipitation of endogenous RAS and SNARE proteins from A549 lysate. **i** Far western blot with spotted purified recombinant protein indicated by row name. Bound recombinant KRAS was detected with an anti-RAS antibody. Coomassie stain for total protein loading (middle) ($n = 3$, representative images shown). **j** Quantification of **i**. Error bars are s.e.m

but did not disrupt SNARE proximity (Fig. 2b, c, d). Deletion of the hypervariable domain essential for membrane localization, however, disrupted KRAS proximity to both Raf1 and SNAREs, suggesting that membrane association sequences are required for both. The prior work on KRAS and SNORD50A/B identified KRAS Lys5, Lys42, Arg149, and Arg161 residues as necessary for SNORD50A/B binding. Mutagenesis of these same residues disrupted SNARE interactions without altering interaction with Raf1 (Fig. 2c, d). SNARE proximity to KRAS therefore depends on the same residues as SNORD50A/B RNAs on an area of the protein independent of effector-binding domains. We next characterized the effect of activating KRAS mutations on SNARE interaction. Given the distinct binding sites of SNARE outside of effector domains, oncogenic mutations in KRAS should not affect association with these SNAREs. In agreement with this, proximity labeling of KRAS$^{WT}$ vs. KRAS$^{G13D}$ reveals no difference in SNARE association between WT and oncomutant KRAS (Fig. 2e, f). Raf1, in contrast, displayed 4-fold increased association with mutant KRAS vs. WT. These data indicate that SNARE proximity to KRAS depends on the same residues as those required for KRAS binding to SNORD50A/B snoRNAs, and these residues are distinct from those involved in effector binding domains.

**SNORD50A/B snoRNAs competitively inhibit KRAS binding with SNAREs.** The nature of the association between KRAS and SNARE proteins was further examined. Super-resolution microscopy revealed that a subset of KRAS protein could be seen in foci suggestive of vesicular localization (Fig. 2g)[33,34]. Consistent with the latter, co-staining with antibodies to SNAP23, SNAP29, and VAMP3 reveals instances of co-localization between KRAS and each of these SNARE proteins, indicating that KRAS and SNAREs can be physically adjacent to each other within the cell. Co-immunoprecipitation of endogenous proteins from cell extracts was next performed. This demonstrated bi-directional detection of KRAS-SNAP23 and SNAP23-VAMP3 protein complexes and uni-directional detection of KRAS-SNAP23, KRAS-SNAP29, KRAS-VAMP3, and SNAP23-SNAP29 complexes (Fig. 2h). Because proximity proteomics does not distinguish direct from indirect interaction, the ability of each SNARE protein to bind KRAS without addition of co-factors was next assessed by far western blotting using purified recombinant proteins. Of the three SNARE proteins, immobilized full length SNAP23 and SNAP29 could capture KRAS from solution; two distinct full length VAMP3 recombinant protein preparations, however, failed to do this, suggesting that VAMP3 proximity to KRAS is indirect (Fig. 2i, j). KRAS, therefore, can be found co-localized with SNAP23, SNAP29, and VAMP3 SNARE proteins within the cell where their endogenous proteins associate with each other, as detected by co-immunoprecipitation. To assess the specificity of direct interaction between these SNAREs and Ras isoforms, we repeated the far western blot this time with recombinant NRAS and HRAS. Again, VAMP3 demonstrated no direct binding activity, but interestingly SNAP29 appeared to have some affinity for all the isoforms (Supplementary Fig. 3A).

SNAP23 appeared to have the greatest specificity for KRAS, consistent with our previous proximity proteomics.

To further confirm and localize the sites of KRAS-SNARE interaction, crosslinking mass spectrometry was next performed between SNAP23/SNAP29 and KRAS. This revealed specific crosslinked residues that overlap the KRAS residues required for RNA binding, supporting direct interaction of these two SNAREs with KRAS (Supplementary Data 2). The requirement for the same KRAS residues for association with both SNORD50A/B snoRNAs and SNAREs and the enrichment of SNARE proteins around KRAS in the absence of these snoRNAs suggested that SNORD50A/B might inhibit SNARE association with KRAS. To explore this, the strength of the interactions between SNORD50A/B RNAs and SNARE interactions with KRAS was first quantitated by microscale thermophoresis (MST)[35, 36]. Purified recombinant KRAS protein bound to in vitro transcribed SNORD50A ($K_d = 140$ nmol/L) and SNORD50B ($K_d = 122$ nmol/L) RNAs whereas the previously identified RNA-binding mutant of KRAS failed to bind either SNORD50A/B in the assayed concentration range (Fig. 3a). The ability of the RNA-binding mutant to load GTP was also assessed by MST and no difference could be observed between the RNA-binding mutant and WT (Supplementary Fig. 3B), indicating that KRAS binding to RNA and GTP are separable processes. Consistent with a model of competitive inhibition, far western competition with SNORD50A/B RNA, but not scrambled control, inhibited KRAS association with SNAP23/SNAP29 (Fig. 3b). Because far western blotting can be quickly saturated and is prone to denaturation-induced artifact, MST was used to quantitate the binding affinity between SNAREs and KRAS. MST demonstrated that KRAS bound purified, recombinant SNAP23 and SNAP29 proteins at $K_d = 177$ nmol/L and $K_d = 178$ nmol/L, respectively, and that binding in the measured concentration range was effectively abolished by SNORD50A/B RNA but not by nucleotide composition and length-matched scrambled RNA control (Fig. 3a, b); full-length Raf1 positive control run in parallel showed a $K_d = 34$ nmol/L, consistent with previously published values for Ras–Raf interactions[37]. Consistent with the previous observation that SNORD50A/B may share a binding surface with KRAS, the RNA-binding mutant also displayed diminished binding affinity to SNAP23 and SNAP29. These data support a model in which SNAP23 and SNAP29 SNARE proteins compete with SNORD50A/B RNAs for binding to KRAS at KRAS residues distinct from those required for binding to GTP.

**SNAREs enable KRAS localization from recycling endosomes to the PM.** SNAP23, SNAP29, and VAMP3 are members of the SNARE vesicular transport family of proteins that reside on membrane and vesicle surfaces (Supplementary Data 3). This raises the possibility that their binding to KRAS may direct KRAS subcellular trafficking and localization within the cell. KRAS trafficking differs from that of its sister isoforms HRAS and NRAS[38,39]. KRAS4B, the major KRAS isoform studied in cancer, contains a C-terminal poly-basic region that mimics lipidation. For KRAS to reach its PM site of active signaling, it must be

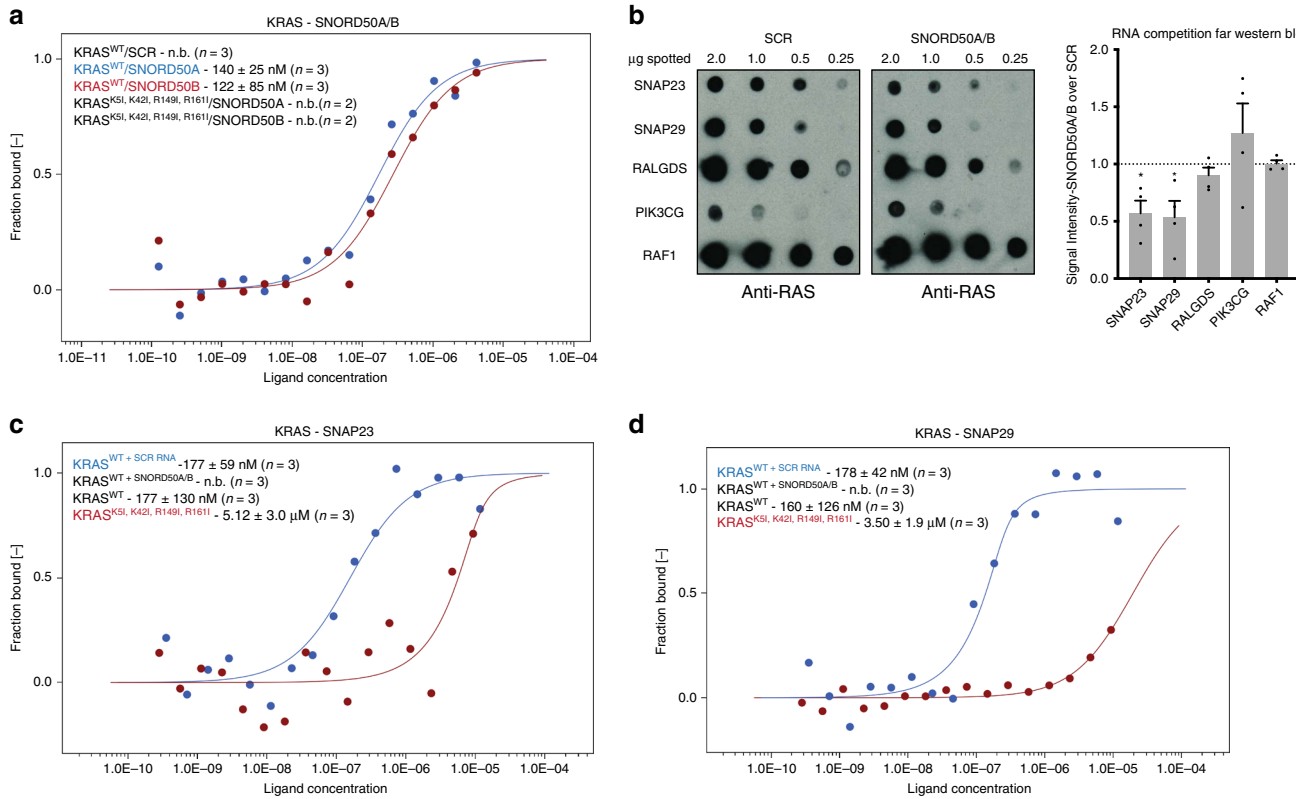

**Fig. 3** SNORD50A/B and SNAREs compete for KRAS binding. **a** Dissociation curves of recombinant purified wild-type KRAS protein and the RNA-binding-deficient KRAS$^{K5,K42R149,R161}$ mutant vs. SNORD50A and SNORD50B snoRNAs measured by MST (n.b. = no binding detected). **b** RNA competition far western of KRAS interaction with SNAREs and canonical effector proteins; SCR = length and sequence nucleotide composition matched scrambled RNA control for SNORD50A/B (n = 3, representative images shown). Significance of SNAP23 and SNAP29 decrease in KRAS association calculated in comparison to Raf1. Dissociation curves of KRAS vs. SNAP23 **c** and SNAP29 **d** by MST in the presence of the RNAs noted. $K_d$ values calculated from 3 independent experiments. Error bars are s.e.m

solubilized from endomembranes by PDEδ[40] and then specifically deposited onto recycling endosomes by ARL2 from where it is trafficked back to the PM[41] To assess the role of SNAP23, SNAP29, and VAMP3 in KRAS localization, we performed global depletion of these protein levels in bulk cell populations using Cas9 and tandem guide RNA sequences targeting each individual gene. Due to the essentiality of SNAP23 to cell survival[42,43], complete ablation was not possible and we proceeded with the best technically achievable disruption of this gene and refer to the bulk depletion of these proteins as SNARE bulk triple knock-out (TKO) (Supplementary Fig. 3C–E, Supplementary Data 4). High-resolution confocal microscopy revealed KRAS localization at the PM with cortical actin in SNARE WT cells (Fig. 4a, Supplementary Fig. 3F). In SNAP29, VAMP3, and SNAP23 TKO displaying globally reduced levels of SNARE proteins, however, KRAS displayed a peri-nuclear concentration co-localized with Arf6 and Rab11 (Fig. 4b, Supplementary Fig. 4A, B), suggesting localization on recycling endosomes. This effect was selective for KRAS as HRAS and NRAS did not change subcellular localization upon SNARE TKO (Fig. 4c, d), consistent with known differences in trafficking of Ras isoforms. In this context, KRAS-bound SNORD50A/B, as measured by ultraviolet crosslinking and KRAS immunoprecipitation (CLIP)[44], is ~8-fold more enriched in membrane compartments than in the cytoplasm (Fig. 4e, f), indicating that KRAS binding to SNORD50A/B occurs in a subcellular space appropriate for SNORD50A/B RNAs to competitively inhibit SNARE binding to KRAS. While the endosomal and vesicular localization of VAMP3 and SNAP29 are well

established, the involvement of SNAP23, a target SNARE thought to primarily function at the plasma membrane, in KRAS trafficking out of the recycling endosomes was surprising. The subcellular localization of SNAP23 was therefore examined by super-resolution microscopy. SNAP23 resides in the expected peripheral distribution, but also co-localizes with Arf6-marked recycling endosomes, indicating that some SNAP23 protein resides in the correct compartment to enable KRAS trafficking (Supplementary Fig. 4C). Loss of SNAP23, SNAP29, and VAMP3 thus leads to KRAS accumulation in the recycling endosome without affecting other Ras isoforms. To explore whether SNARE depletion or SNORD50A/B overexpression altered the interaction of KRAS with PDEd, PLA was undertaken and demonstrated no significant differences in interaction rates under these conditions, suggesting the effects were independent of cytoplasmic solubilization (Supplementary Fig. 4D–F). Importantly, the two KRAS variants, KRAS4A and KRAS4B, have been described to both be expressed to varying degrees in human cancer tissue, and differ in their interaction with the PDEδ solubilization factor. We therefore explored whether KRAS4A was also divergent with KRAS4B in this vesicular transport axis. Both KRAS4A and KRAS4B fail to rescue the survival effect of SNARE disruption, suggesting that the trafficking of both may converge on this SNARE/SNORD50A/B − regulated pathway, consistent with the previous observation that both KRAS4A and KRAS4B bind SNORD50A/B (Supplementary Fig. 4G, H)[16]. Further work is necessary to fully elucidate the differences between KRAS4A and KRAS4B sub-cellular transport.

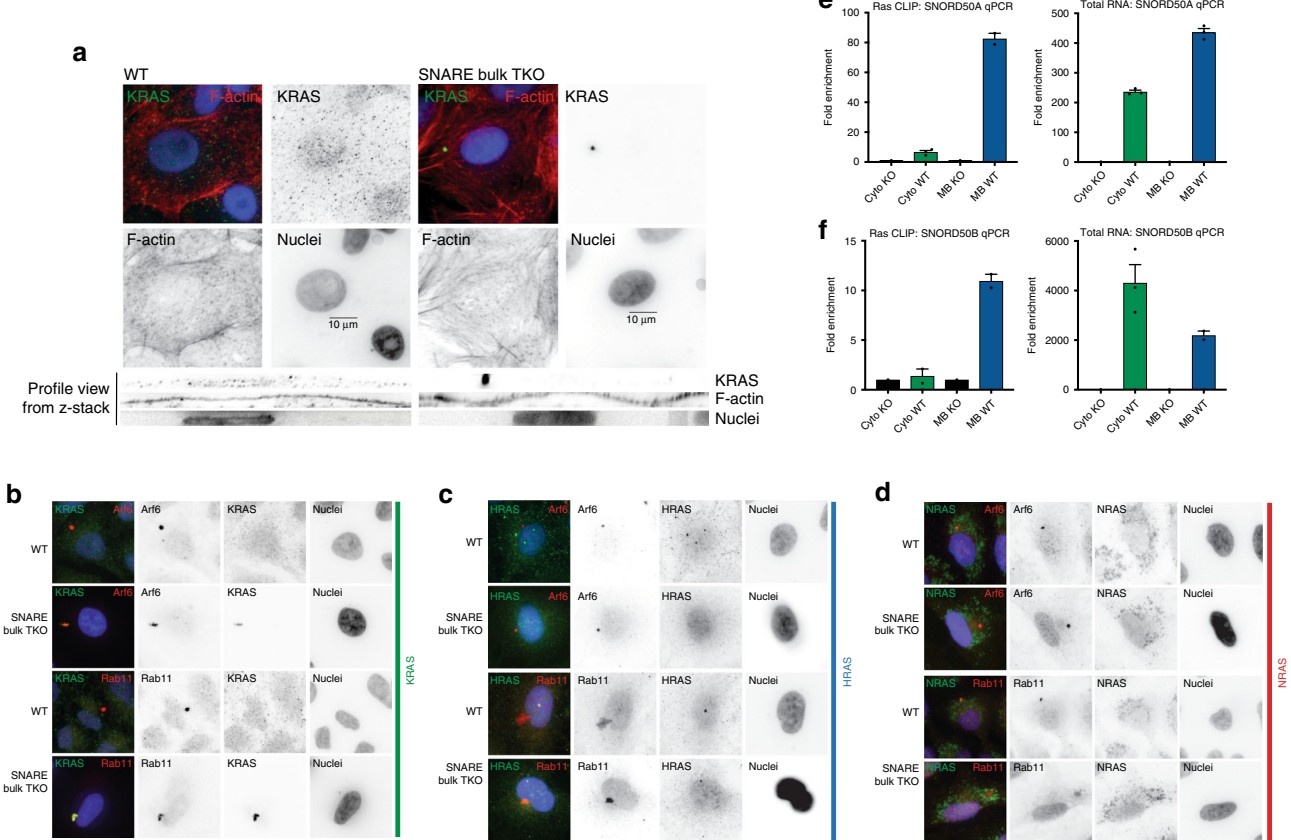

**Fig. 4** SNARE deletion leads to KRAS localization in the recycling endosome. **a** Immunofluorescence of endogenous KRAS in A549 cells. Side profile view of each cell (below); note KRAS localization at the PM with cortical actin and its re-localization to a perinuclear focus with SNARE TKO. Scale bars are 10 μm. **b** KRAS colocalization with the recycling endosome markers, Arf6 and Rab11 as a function of SNARE TKO in A549. **c** HRAS and **d** NRAS colocalization with Arf6 and Rab11 as a function of SNARE TKO in A549. **e** CLIP-qPCR against SNORD50A and **f** SNORD50B after immunoprecipitation of endogenous RAS in either cytoplasmic or membrane bound subcellular fractions (left) from H23. qPCR against SNORD50A and SNORD50B in total RNA isolated from cytoplasmic or membrane bound fractions (right) from *SNORD50A/B* WT or KO cells from H23. Error bars are s.e.m

**SNARE proteins are essential for downstream KRAS signaling.** Compromising KRAS arrival at the PM should impede its ability to associate with its receptors and downstream effectors. Consistent with this, KRAS proximity to EGFR and its canonical downstream kinases, Raf1 and PI3K P110α decreased with SNARE TKO and increased with SNORD50A/B KO in cancer cells with activating oncogenic KRAS mutations (Fig. 5a, b, c). Interestingly, while the interaction of KRAS with receptors and effectors is altered, SNARE KO and SNORD50A/B did not appear to significantly decrease the formation of GTP-Ras with exogenously expressed WT KRAS, suggesting that some basal interactions with GEFs may still occur in endosomes consistent with previous reports (Supplementary Fig. 4I, J)[45–47]. Furthermore, SNARE-SNORD50A/B antagonism could be seen in signaling pathways downstream of KRAS. SNARE TKO reduced levels of active ERK and AKT driven by oncogenic KRAS while SNORD50A/B KO increased them (Fig. 5d; Supplementary Fig. 4K). The effects of SNARE ablation on KRAS signaling could also be seen on KRAS target gene regulation. RNA sequencing (Fig. 5e, Supplementary Data 5) demonstrated that SNARE knockout downregulated the published KRAS-dependent transcriptional signature[48], as well as proliferation genes and previously identified oncogenic signatures (Fig. 5f). The ability of KRAS to interact with upstream EGFR, as well as downstream Raf1 and PI3K P110α effector kinases therefore depends on SNAREs, as does KRAS impacts on gene regulation.

**SNAREs are a vulnerability of KRAS-driven cancers.** The requirement for specific SNARE proteins for KRAS localization, signaling and gene regulation, suggests these SNAREs might play a role in KRAS-driven cancer. Consistent with this, analysis of TCGA[49,50] patient outcomes from pancreatic cancer, the type most frequently associated with oncogenic mutations in the *KRAS* gene, demonstrated that patients with mutated or down-regulated SNAP23 or VAMP3 survive longer than counterparts in whom these SNAREs are uncompromised (Fig. 6a, b; Supplementary Data 6). Additionally, KRAS mutant tumors displayed higher levels of SNAP23 and VAMP3 expression (Supplementary Fig. 5A, B) in line with a synergistic model of KRAS-signaling with SNARE enablers of KRAS transport to the PM. Analysis of additional KRAS-driven tumor types (Supplementary Fig. 5C–E) demonstrated a similar initial survival trend in KRAS mutant colon cancer, where KRAS was also readily observed proximal to these SNAREs in a series of patient tumor specimens (Supplementary Fig. 5F, G). These data are consistent with a potential role for KRAS-interacting SNARE proteins in human cancer.

To test this experimentally, the impact of individual SNAP23, SNAP29, and VAMP3 KOs in KRAS-driven tumorigenesis was assessed. Similar to the association pattern seen in TCGA survival data, in vivo tumorigenesis by KRAS-dependent human tumor cells in immune deficient mice required SNAP23 and VAMP3, but not SNAP29 (Fig. 6c; Supplementary Fig. 6A). This along with the previously observed outcomes in pancreatic cancer patients may suggest a more redundant role for SNAP29.

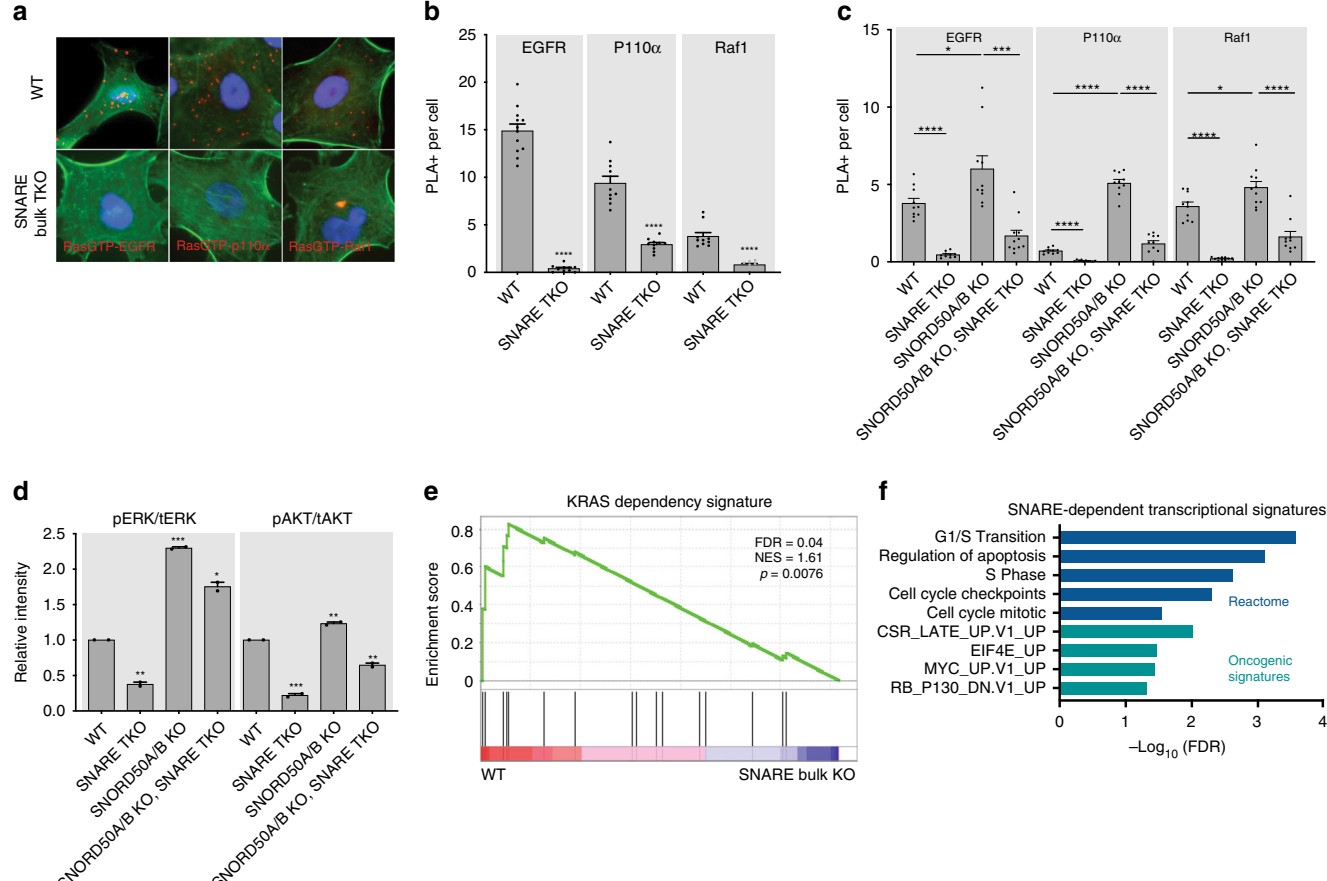

**Fig. 5** Downstream KRAS signaling relies on SNARE proteins. **a** Representative PLA images of Ras-GTP interaction with EGFR, P110α, and Raf1 in A549. F-actin labeled in green. **b** Quantification of PLA signal in **a**. **c** Proximity of KRAS with membrane bound receptors and signaling effector proteins measured by PLA in H23 cells. **d** ERK1/2 and AKT activation in H23 cells; quantitated replicate western blots of phosphorylated pERK and pAKT over total levels of these kinases (tERK, tAkt) are shown. **e** GSEA plot of KRAS-dependent transcripts against RNA-sequencing datasets of SNARE wild-type and KO cells in H23. **f** Gene sets lost upon SNARE TKO in H23, A549, and CHL1 from RNA sequencing. Error bars are s.e.m

To distinguish KRAS-selective effects versus global impacts on Ras-signaling and tumorigenesis by SNARE loss, DLD-1 isogenic colon cancer subclones[51], which differ only by whether they express a single WT or oncogenic point mutant KRAS allele, were next used. SNARE TKO reduced active AKT and ERK signaling in xenografted tumors driven by oncogenic KRAS and reduced tumor size by the oncogenic KRAS mutant subclone down to that seen with the KRAS WT isogenic line, which still formed tumors of comparable size whether SNAREs were knocked out or not (Fig. 6d–g; Supplementary Fig. 6B). This suggests that these SNAREs are required for KRAS-augmented, but not global, tumorigenic capacity. DLD-1 has been previously reported to be selectively sensitive to KRAS signaling disruption in tumors and anchorage – independent growth but minimally in culture. We therefore assessed and observed this KRAS-selective inhibition in an otherwise identical genetic background with SNARE TKO in anchorage independent growth but not growth in 2-dimensional culture (Supplementary Fig. 6C, D), indicating that SNARE reduction inhibits KRAS-enabled tumorigenesis features without impairing overall cell viability. SNORD50A/B overexpression was then performed in DLD-1, demonstrating the expected reciprocal result of suppressing ERK activation (Supplementary Fig. 6E).

To further address the necessity of SNAREs in KRAS-driven tumorigenesis, as well as their specificity for KRAS over other Ras isoforms, we performed a CRISPR correlated gene essentiality screen[52]. Tumorigenesis of 11 independent human cancer lines

(A549, MM485, BxPC3, AsPC1, LS174T, HPAFII, HCT116, H460, H358, CHL1, DLD-1) in vivo, along with 3 subclones (CHL1 SNORD50A/B KO, DLD-1 KRASWT/-, DLD-1, KRASG13D/-), was studied with 1000 guide RNAs to assess the functional link between KRAS, HRAS, NRAS, and SNAREs in tumorigenesis (Fig. 7a). Using co-essentiality analysis, where the phenotypic correlation of genes can be assessed based on depletion or enrichment in multiple cell lines, SNAP23 was found to correlate strongly with KRAS but not with HRAS or NRAS (Fig. 7b; Supplementary Fig. 6F; Supplementary Data 7). A complementary analysis of co-essentiality in 340 cancer cell lines in which whole genome CRIPSR KO screening has been performed in the Project Achilles effort also revealed significant and specific SNAP23 – KRAS correlation, suggesting that SNAP23's functions are especially critical in enabling tumorigenesis driven by KRAS, but not other Ras isoforms (Supplementary Fig. 6G).

Given the repeated evidence of the strongest functional relationship between SNAP23 and KRAS, the mechanism of KRAS relocalization was revisited with the hypothesis that SNAP23 may be the most critical function interaction. Indeed, repeated immunofluorescence in A549 reveals the greatest phenocopy of recycling endosome KRAS restriction in SNAP23 KO (Supplementary Fig. 7A). This appears to be predominantly a shift from the plasma membrane compartment to the recycling endosome membrane, as the cytoplasmic/membrane ratio of KRAS was not significantly changed with SNARE KOs

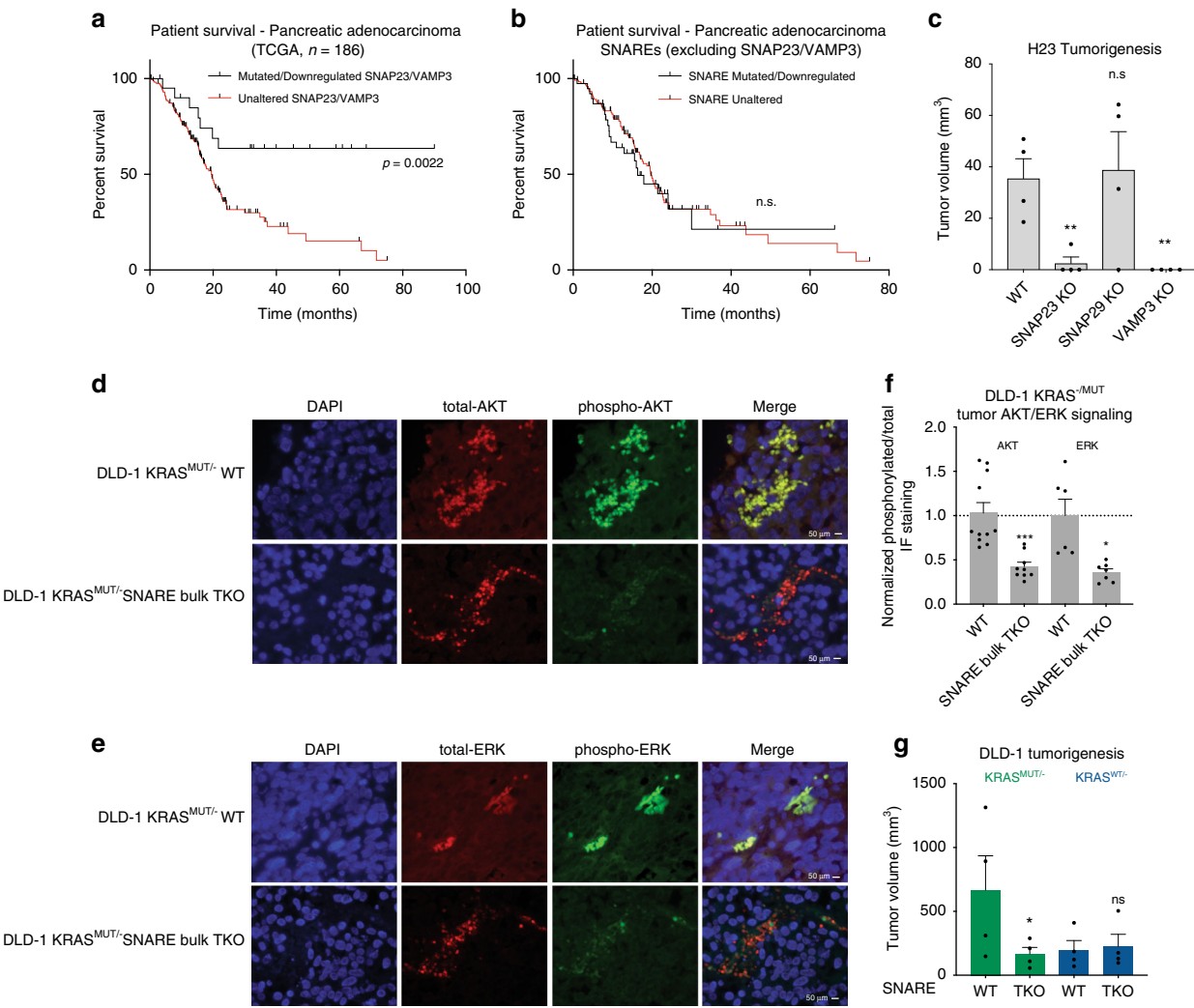

**Fig. 6** SNARE-enabled transport is a vulnerability of KRAS-driven cancer. **a** Survival of KRAS-mutant pancreatic adenocarcinoma patients depending on mutation and expression status of SNAP23 and VAMP3 ($n = 186$). **b** Survival of KRAS-mutant pancreatic adenocarcinoma patients depending on mutation and expression status of the 36 other SNARE proteins ($n = 162$). **c** Subcutaneous in vivo tumor growth of H23 SHO immune deficient mice ($n = 4$ mice/group). **d** Immunofluorescence images of phosphorylated and total AKT in oncogenic KRAS-driven DLD-1 subcutaneous tumors. **e** Immunofluorescence images of phosphorylated and total ERK in oncogenic KRAS-driven DLD-1 subcutaneous tumors. **f** Quantitation of **d** and **e**. **g** Subcutaneous in vivo tumor growth in DLD-1 isogenic lines ($n = 4$ mice/group). Error bars are s.e.m

(Supplementary Fig. 7B, C). To further clarify the role of SNAP23 on KRAS localization, SNORD50A/B overexpression was undertaken in A549. In support of the previously proposed competitive inhibition, SNORD50A/B results in partial sequestration of KRAS into recycling endosomes (Supplementary Fig. 8A). Additionally, SNORD50A/B can be directly detected in the recycling endosome in close proximity to KRAS (Supplementary Fig. 8B). Using super-resolution microscopy, the co-localization of SNAP23 and KRAS in this compartment was compared. As earlier noted in WT conditions, KRAS and SNAP23 strongly co-localize. In the context SNORD50A/B overexpression, however, there is greater spatial separation of the two proteins on recycling endosomes, an observation not found in SNORD33 or SNORD83B overexpression – two snoRNAs confirmed to have no meaningful KRAS binding activity (Supplementary Data 8–11, Supplementary Fig. 8C–E)[16]. We hypothesize that direct interaction with SNAP23 may be critical to recruit KRAS to an actively transported subdomain of the recycling endosome.

The SNARE requirement for KRAS-driven tumorigenesis suggested that targeting specific SNAREs by methods orthogonal to genetics might also alter this process. Among KRAS-proximal

SNAREs, SNAP23 was chosen for targeting because it binds KRAS directly, its loss is associated with better patient survival in TCGA data, and because its single and combined KO abolish KRAS-driven experimental tumorigenesis. To do this, protease toxins from C. botulinum and C. tetani, which cleave SNARE proteins, were examined. Botulism toxin isoform A is used in disorders of overactive cholinergic nerve terminals, indicating that these toxins can be safely used as therapeutics[53]. To target SNAP23, we used the catalytic light chain K224D mutant of the botulism toxin protease isoform E first described and characterized by Chen and Barbieri[54], which to our knowledge is the only described, engineered isoform to degrade human SNAP23, and compared it to the non-mutant protease, which degrades the neural tissue-localized SNARE, SNAP25[54–56] (Supplementary Fig. 9A–D). Expression of the K224D mutant protease, but not the WT protease control, decreased KRAS-driven in vivo tumorigenesis (Fig. 7c, d). Targeting KRAS-interacting SNAREs with a SNAP23-degrading protease thus also impairs tumorigenesis, further indicating that targeting SNARE proteins may represent a alternative therapeutic opportunity in KRAS-driven cancer.

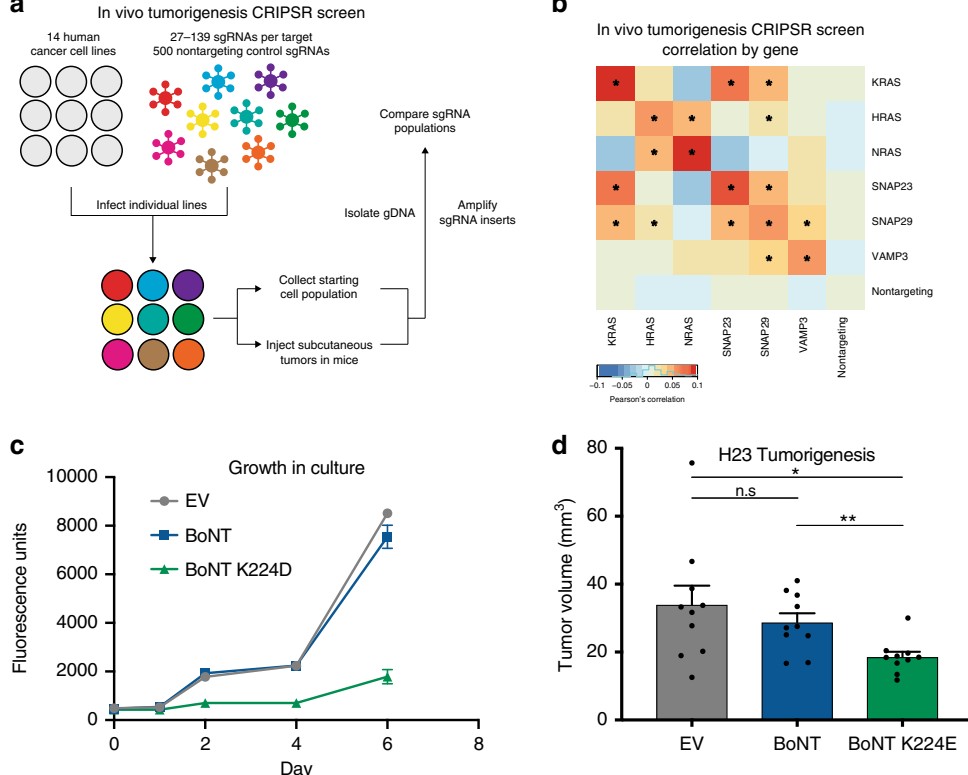

**Fig. 7** SNAP23 is a potentially targetable vulnerability of KRAS. **a** Schematic of a focused co-essentiality CRISPR screen targeting KRAS, HRAS, NRAS, SNAP23, SNAP29, and VAMP3 ($n = 14$ tumors). **b** Correlation of in vivo tumorigenesis sgRNA counts by target gene in CRISPR screen. **c** Growth in culture of H23 expressing BoNT light chain E (LC/E) and LC/E$^{K224D}$ ($n = 2$). **d** Tumor volumes of H23 expressing BoNT light chain E (LC/E) and LC/E$^{K224D}$ at day 16 ($n = 10$ tumors per condition). Error bars are s.e.m

## Discussion

Here we identify a KRAS regulatory mechanism in which specific SNARE proteins enable KRAS arrival at the PM in a process opposed by SNORD50A/B RNAs. Altering the balance of antagonistic KRAS regulation by SNARE-SNORD50A/B changes KRAS signaling, target gene regulation, and in vivo tumorigenesis. These findings support a revised working model of KRAS transport to the PM that involves SNAP23, SNAP29, and VAMP3. SNAP23 appears to play an especially critical and non-redundant role in allowing KRAS to drive tumorigenesis. This model posits that these SNARE proteins are required for KRAS trafficking out of the recycling endosomes to reach the PM where KRAS can interact with upstream receptors and downstream effectors. It proposes that SNARE-enabled KRAS arrival at the PM is antagonized by SNORD50A/B snoRNAs, which compete with SNAREs for binding to KRAS. This model suggests that the frequent deletion of *SNORD50A/B* observed in cancer, which is enriched for co-occurrence with oncogenic *KRAS* mutations[16], removes a block to SNARE-facilitated KRAS trafficking to the PM where it can mediate its oncogenic signaling functions.

These data point to a role for SNARE proteins in Ras regulation. Since the discovery of PDEδ-mediated KRAS solubilization, KRAS recycling and trafficking has come to attention as a potential cancer target, with recent efforts devoted to targeting PDEδ. The present results suggest a mechanism by which SNAREs, known to reside on endosome and vesicle surfaces, bind KRAS to direct it to endosome compartments destined for the PM, which may represent an essential step to maintaining KRAS enrichment at its site of active signaling. SNAREs' primary known role is to fuse vesicles with target membranes to facilitate the transfer of vesicle-bound cargo. SNAREs, however, may also bind

KRAS and other proteins as vesicle surface cargo. This raises the possibility that vesicular transport to the PM in this context serves two purposes: (1) exocytosis of vesicle-containing material (2) directed protein transport from the surface of endosomes to the inner leaflet of the PM.

To our knowledge, this is the first report of snoRNAs binding a protein to influence its subcellular trafficking, congruent with expanding roles of snoRNAs outside of canonical ribosomal RNA modification. Additionally, the observation that KRAS binds non-coding RNA raises the possibility that GTPase interactions with snoRNAs may represent a conserved class of interactions. In support of this, the Bms1 ribosomal GTPase binds U3 snoRNA in 40S synthesis, and 23S rRNA can control the function of ribosomal GTPases[57–59]. These observations provide a rationale for future efforts to systematically characterize the spectrum of RNA binding capacity of Ras superfamily GTPases.

KRAS is a driver of many highly aggressive human malignancies including pancreatic, colon, and lung cancers. Although recent efforts have made promising progress towards small molecule inhibition of Ras, no effective therapy is currently available. Targeting the receptors upstream of Ras isoforms, such as EGFR, as well as its downstream effectors, such as BRAF, however, has proven to be a successful strategy to achieve positive clinical impacts in human cancer. Elucidating the details of the KRAS protein's lifecycle is therefore of significant interest, not only in understanding mechanisms of oncogene biology, but also in informing therapeutic strategies towards directed at KRAS-interacting proteins essential for KRAS trafficking and oncogenesis.

In this regard, the dependence of KRAS on SNARE-directed vesicular transport opens up possibilities for targeting KRAS-driven tumors by exploiting SNARE-cleaving botulism toxins. Findings

here demonstrate that altering a botulism toxin protease to expand its activity to cleave SNAP23 can suppress tumorigenesis. Given the modular nature of the receptor-binding heavy chain and enzymatic light chain of botulism toxin, these proteases may be further engineered for target cell specificity and SNAP23-specificity[14]. Such a SNARE-based approach to impairing KRAS arrival at the PM could be synergistic with other approaches to drugging Ras proteins directly[60,61]. These data thus suggest that small non-coding RNAs and bacterial toxins may expand the arsenal with which to modulate KRAS and provide a rationale for future efforts to target specific SNARE proteins in KRAS-driven cancer.

## Methods

**Cell Culture**. All cell lines were growth with 10%FBS and 1% pen-Strep at 37 °C, 5% $CO_2$. H23, A549, CHL1 and their respective subcloned lines were cultured in DMEM. H358 and H460 were cultured in RPMI. HCT116, DLD-1 and its respective isogenic lines were cultured in McCoy's 5A. H23s, A549s, and CHL1s were purchased from ATCC, and DLD-1 s were purchased from Horizon. All other cell lines including MM485, BxPC3, AsPC1, LS174T, HPAFII, and HCT116 were cultured in their recommended media from ATCC. All cell lines were mycoplasma free by the MycoAlert mycoplasma detection kit (Lonza). Information about the gender and tissue source of the cell lines are available in the Key Resources table.

For experiments investigating the role of SNAREs in KRAS biology, H23 and A549 were selected from the original three cell lines (CHL1, H23, and A549) because they are driven by at least one oncogenic KRAS allele. Tumorigenesis, survival, and signaling experiments were performed primarily in H23 because H23 is sensitive to KRAS KD while A549 are less sensitive. We observed a 71% reduction in H23 cellular proliferation ($n = 2$) and 34% reduction in A549 cellular proliferation ($n = 2$) by cell titer blue assay consistent with previous reports[48,62]. Select critical experiments were performed in both cell lines.

**Lentivirus Production and Infection**. Lentivirus was produced by transfecting 293T s with 10 μg pLex lentiviral expression construct, 7.5 μg pCMVΔ8.91, and 2.5 μg pUC MDG using Transit X2 (Miuras) in a 10 cm plate. Virus was harvested 48 and 72 h after transfection and filtered through 0.45 μm filters to remove cell debris and concentrated using Lenti-X concentrator (Clontech). All cell lines were transduced overnight in 5 μg/mL polybrene and selected with 1 μg/mL puromycin or blasticidin when applicable. Cells were selected for a minimum of 2 days for protein expression and 5 days for Cas9-mediated gene editing.

**Proximity Protein Labeling/Mass Spectrometry**. Fusion constructs for vicinal protein labeling were generated in pLex lentiviral expression constructs by expression HA-BirA on the N terminal of KRAS connected by a 10 amino acid Glycine/serine linker. KRAS mutants were generated as previously described in Siprashvili et al. 2016[16].

H23, A549, and CHL1 parental cells lines as well as two subcloned lines with homozygous SNORD50A/B knockout were transduced with pLex-HA-BirA-KRAS. At ~80 confluence, media was supplemented with biotin to a final concentration of 50 μM. Cell were harvested 24 h later[63] with the addition of a filtration step through a 3 K MWCO column prior to MyOne C1 bead binding. For western blotting, 1 mg of total protein input was eluted in 30ul of elution buffer. For mass spectrometry, an input of ~10 mgs of total protein was used and half of the eluted product was run on a 4–12% Bis-Tris SDS-PAGE gel and stained with Colloidal Blue (ThermoFischer). A single 1 $cm^2$ gel slice per sample was fixed in 50% methanol/10% Acetic acid then stored in 1% acetic acid. Each gel was then digested as previously described[64]. Isolated peptides were then reconstituted and injected into a 25 cm C18 reversed phase analytical column in a Waters NanoAcquity run at 450 nL/min from a 4 to 35% mobile phase (0.1% formic acid). An Orbitrap Elite was set to acquire data selecting and fragmenting 15 precursor ions with the greatest intensity in the ion-trap where the exclusion window was set at 45 s and multiple charge states allowed.

Lentiviral expression constructs were additionally generated for HRAS and NRAS in the same manner as above in order to determine relative SNARE interaction strengths. Each isoforms was expressed in a cancer cell type frequency associated with activating mutations in the respective RAS isoform. BirA-KRAS was expressed in CaCO2, BirA-HRAS was expressed in HT1376, and BirA-NRAS was expressed in CHL1. In each cell line, BirA was also expressed alone to generate cell-line specific background controls.

MS/MS data were analyzed using both Preview and Byonic v2.6.49 (ProteinMetrics). All data were first analyzed in Preview to provide recalibration criteria if necessary and then reformatted to MGF before full analysis with Byonic. Analyses used Uniprot canonical and isoform FASTA files for Human with mutant sequences concatenated as well as common contaminant proteins. Data were searched at 12 ppm mass tolerances for precursors, with 0.4 Da fragment mass tolerances assuming up to two missed cleavages and allowing for fully specific and N-ragged tryptic digestion. These data were validated at a 1% false discovery rate using typical reverse-decoy techniques (Elias Nat. Meth. 2007). The resulting

identified peptide spectral matches and assigned proteins were then exported for further analysis using custom tools developed in MatLab (MathWorks) to provide visualization and statistical characterization.

Raw spectral counts were collapsed by gene name and probability scores were calculated for each bait-prey combination using SAINT analysis (crapome.org) using the following parameters: 10,000 iterations, LowMode ON, Normalize ON and the union of MinFold ON and OFF. Available birA controls in crapome.org as well internal controls were used as background. To identify SNORD50A/B-modulated interactions, probability score change of 0.25 between SNORD50A/B WT and KO datasets and a raw spectral count fold change of >1.5 in at least two of three cell lines were required.

Protein network diagrams were generated in Cytoscape running GeneMANIA to identify protein–protein interactions, pathways, and enrichment scores. Total network connectivity and GO terms for SNORD50A/B-modulated interactions were calculated from the STRING database.

**Crosslinking mass spectrometry**. Crosslinking proteins was performed using $BS^3$ as previously described in Schmidt et al.[15]. Crosslinked reactions were then run on a 4–12% SDS-PAGE gel and resolved by colloidal staining. Gel slices were cut in 1 × 1 mm squares above the point in the gel where a reference sample where no crosslinking agent had been added. The excised gel bands were then rinsed multiple times with 50 mM ammonium bicarbonate and reduced with 5 mM DTT, 50 mM ammonium bicarbonate at 55 °C for 30 min. Residual solvent was removed and alkylation was performed using 10 mM propionamide in 50 mM ammonium bicarbonate for 30 min at room temperature. The gel pieces were rinsed 3 times with 50% acetonitrile, 50 mM ammonium bicarbonate and placed in a speed vac for 5 min. Digestion was performed with trypsin/LysC (Promega) in both a standard overnight digest (14 h) at 37 °C, as well as in a limited digest format (1 h at 50 °C). Tubes were centrifuged and the solvent including peptides were collected and further peptide extraction was performed by the addition of 60% acetonitrile, 39.9% water, 0.1% formic acid and incubation for 10–15 min. The peptide pools were dried in a speed vac.

Digested cross-linked peptide pools were reconstituted and injected onto a 100 micron I.D. C18 reversed phase analytical column, 25–50 cm in length. The UPLC was a Waters M class, operated at 300 nL/min using a linear gradient from 4% to 35% mobile phase B. Mobile phase A consisted of 0.1% formic acid, 5% DMSO; mobile phase B was 0.1% formic acid, 5% DMSO, acetonitrile. All data were collected using an Orbitrap Fusion mass spectrometer set to acquire data in a data dependent decision tree fashion selecting and fragmenting by ETD or HCD the most intense precursor ions defined by a predetermined schema where both $m/z$ and charge state are considered. An exclusion window of 60 s was used to improve proteomic depth and multiple charge states of the same ion were sampled.

All MS/MS data were analyzed using Preview, Byonic v2.6.49 and Byologic v. 2.6–73 (ProteinMetrics) as well as custom in house tools for data analysis developed in MATLAB. Preview was used to verify system calibration using uncrosslinked peptides prior to Byonic analysis. For $BS^3$ crosslinks, Byonic used the exact mass of peptide fragments for crosslink assignment. For zero-length crosslinking, Byonic analyses were completed using the 'Xlink' functionality to generate a complete list of possible crosslinked peptides using a custom FASTA file containing the target protein sequences. For both analyses, sequences were searched with a reverse-decoy strategy at a 1% false discovery rate to identify both crosslinked and uncrosslinked peptide assignments. Byonic searches were performed using 10 ppm mass tolerances for precursor and HCD fragment ions, and with 0.3 Da tolerances for ETD fragment ions detected in the ion trap. In addition, these searches required fully specific tryptic or chymotryptic digestion allowing up to two missed cleavages per peptide. The resulting identified potential crosslinked peptide spectral matches were then exported for further analysis by Byologic. Crosslinked spectra were required to meet the following criteria: 1) all peptides, crosslinked or native, were filtered to a <1% FDR; 2) precursor mass error of less than 7 ppm was required for crosslinked peptides. Positive hits were filtered through a common contaminants database[65].

**Far Western Blotting**. The indicated amounts of recombinant GST(Sigma-Aldrich), SNAP25(abcam), SNAP23(Origene), SNAP29(Origene), VAMP3 (LSbiosciences), VAMP3(MyBiosource), RALGDS(Origene), PIK3CG(Origene), or Raf1(Origene) were spotted on a nitrocellulose membrane and allowed to dry for 30 min. The membrane was then blocked in 5% milk in TBST and incubated overnight at 4 C with 1 μg KRAS protein (abcam). For RNA competition far westerns, in vitro transcribed RNA was preincubated with 1 μg KRAS protein at 4 C for 1 h prior to overnight incubation with the membrane. The membrane was then washed in 5% milk in TBST and incubated with horseradish peroxidase (HRP)-conjugated secondary antibody for 1 h at RT. The membrane was then developed using the SuperSignal West Dura reagent (ThermoFischer). Quantification was done using ImageJ. For loading controls, duplicate membranes were spotted and immediately stained with a Coomassie Stain Kit (Biorad).

**In Vitro Transcription and Protein Microarray Hybridization**. SNORD33, SNORD83B, SNORD50A, SNORD50B, and a scrambled sequence based off of SNORD50A/B were transcribed from pCr2.1 TOPO vectors as previously

described[1]. RNAs were then folded in TE buffer by heating to 70 °C for 5 min and immediately transferring on ice for 15 min. Protein microarray hybridization and analysis were performed as specified in Siprashvili et al., 2016[16].

**SnoRNA Overexpression**. SNORD50A, SNORD50B, SNORD33 and SNORD83B were synthesized as IDT gBLOCKS and in-fusion cloned into pSPARTA snoRNA production vector seamlessly[66] allowing snoRNA expression from the intron of the SNHG5 gene. A mock infection plasmid was generated in a similar manner with no inserted sequence.

**Microscale thermophoresis**. Target proteins were labeled with cysteine-reactive dye, NT-647-Red-NHS (Nanotemper Technologies) in PBS buffer at three molar express to protein solution buffer exchanged into PBS. The protein-dye mixture was incubated at 4 C for 45 min and purified on a desalting column equilibrated with PBS. Binding affinities were analyzed on a Monolith NT.115 instrument (Nanotemper Technologies). In total 16-point dilutions of indicated ligands were prepared in either PBS (for proteins) or deionized water (for RNA) and mixed 1:1 with labeled target solution so that the final target solution was 20 nM. Reaction mixtures were run on "medium" power and 15% LED power. The dissociation constant Ks is calculated by fitting the binding curve with the quadratic solution for the fraction of fluorescent molecules that formed the complex, calculated from the law of mass action: $[AT] = 1/2*(([A0] + [T0] + Kd)-(([A0] + [T0] + Kd)2-4*[A0]*[T0])1/2)$ where $[A]$ is the concentration of free fluorescent molecule, $[T]$ the concentration of free titrant and $[AT]$ is the concentration of complexes of $[A]$ and $[T]$, $[A0]$ is the known concentration of the fluorescent molecule and $[T0]$ is the known concentration of added titrant. In RNA competition assays, RNA was in vitro transcribed as previous described. In each reaction, either an equimolar mixture of SNORD50A and SNORD50B or a length and composition matched scrambled RNA control was added to the labeled target to a final fixed concentration of 4.15 nM while the ligand was titrated in 16, 2-fold intervals. Noise, signal to noise ratio, and response amplitude were measure and acceptable threshold values were analyzed in MO.Control v1.6.1. Dissociation curves that did not meet threshold values were not plotted and designated as 'no binding', meaning no detectable fractional binding shift.

**Immunofluorescence and Proximity Ligation Assay**. Cells were fixed with 4% formaldehyde at RT for 15 min then washed twice with PBS. Blocking was done either for 1 h at RT or overnight at 4 C in blocking buffer (5% goat serum, 5% horse serum, 0.3% TritonX-100). Primary antibodies were incubated overnight at 4 C in blocking buffer and secondary antibodies were incubated for 1 hr at RT. Slides were then mounted in Prolong Gold with DAPI (Invitrogen). Primary antibodies used were as follows: anti-Panras C-4 (sc-166691, Santa Cruz Biotechnology), anti-RASGTP (26909, Neweast Biosciences) (1:250 dilution), anti-KRAS (12063-1-AP, Proteintech), anti-HRAS (18295-1-AP, Proteintech), anti-NRAS (10724-1-AP, Proteintech), anti-SNAP23 (10825-1-AP, Proteintech), anti-SNAP29 (12704-1-AP, Proteintech), anti-VAMP3 (10702-1-AP, Proteintech), anti-Raf1 C12 (sc-133, Santa Cruz Biotechnology), anti-EGFR 1005 (sc-03-G, Santa Cruz Biotechnology), anti-p110α (4249, Cell Signaling Technologies), anti-RAB11 KT80 (ab95375, Abcam), anti-ARF6 3A-1 (sc-7971, Santa Cruz Biotechnology), anti-Phospho-p44/42 MAPK (Erk1/2) (Thr202/Tyr204) (4377, Cell Signaling Technologies), anti-p44/42 Erk1/2 (9107, Cell Signaling Technologies), anti-Phospho-AKT (Ser473) (4060, Cell Signaling Technologies), anti-AKT(pan) (2920, Cell Signaling Technologies). Secondary Antibodies used were as follows: anti-Rabbit Alexa Fluor 555 (ThermoFischer Scientific), anti-Mouse Alexa Fluor 555 (ThermoFischer Scientific), anti-Rabbit Alexa Fluor 488 (Thermo Fischer Scientific), and anti-Mouse Alexa Fluor 488 (ThermoFischer Scientific). Antisense oligos were ordered from Integrated DNA technologies with a 5′ biotin modification. These were incubated under the same conditions for antibodies and detected with the appropriate streptavidin-linked fluorophore conjugate (Thermo Scientific). Unless otherwise noted, all primary antibodies were used at a dilution of 1:1000 and all secondary antibodies used at a dilution of 1:250.

For PLA, Duolink mouse minus, rabbit plus, and In Situ Orange Reagents (Sigma-Aldrich) were used according to the manufacturer's protocol. When applicable, a cytoskeletal stain using pre-conjugated 488 phalloidin was used (Invitrogen) after polymerase reaction and prior to slide mounting. Images were analyzed using ImageJ.

For imaging of xenografted tumors excised from mice subcutaneous growths, tumors were fixed in formalin and briefly stored in 70% ethanol. Following paraffin-embedding and sectioning, antigen retrieval was performed as follows: (1) Two washes in xylene (5 min each) (2) Two washes in 100% ethanol (5 min each) (3) Two washes in 95% ethanol (5 min each) (4) Two washes in 70% ethanol (5 min each) (5) Two washes in dH2O ethanol (5 min each) (6) Slides brought to a boil in citric acid based buffer (VectroLab #H3300) (7) Cooled for 30–60 min (8) Washed in PBS for 5 min. Staining was then performed as described above.

For patient tumor samples (Asterand Biosciences), prior to blocking, formalin-fixed, paraffin-embedded samples were rehydrated in xylene and sequential dilutions in ethanol followed by Citrate Buffer antigen retrieval as described above.

**Microscopy**. Images were obtained either on a Nikon spinning disk confocal microscope or an OMX Blaze 3D-structured illumination, super-resolution microscope (Applied Precision, Inc.). Image analysis was performed with a combination of ImageJ, Matlab, and softWoRx Imaging Workstation.

**Crosslinking followed by Immunoprecipitation**. H23 and a SNORD50A/B KO subclone of the parental line were subjected to 0.3 J/cm² UV-C crosslinking on ice then subjected to fractionation using the Plasma Membrane Isolation Kit (Abcam) according to manufacturer's instructions. A fraction of each sample was set aside for total RNA isolation and another was resuspended 5 volumes CLIP lysis buffer (50 mM Tris, pH 7.5, 10% glycerol, 200 mM NaCl, 5 mM EDTA, 0.5% sarkosyl, 0.2% Tween and 0.1% Igepal). The fraction set aside for CLIP was incubated overnight at 4 C with end over end turning with Protein G beads (ThermoFischer Scientific) precoupled to an anti-panras C4 antibody. Proteinase K (New England Biosciences) was then added to release RNA. At this time both total RNA as well as CLIP fractions were purified using the miRNeasy Mini Kit (Qiagen). Reverse transcription using the ISCRIPT cDNA synthesis kit (Biorad) with target specific priming for SNORD50A/B was used to generate cDNA, and qPCR was performed using SYBRgreen (ThermoFischer Scientific).

**Immunoblot Assays**. The following antibodies were used: anti-Panras C-4 (sc-166691, Santa Cruz Biotechnology), anti-RASGTP (26909, Neweast Biosciences), anti-SNAP23 (10825-1-AP, Proteintech), anti-SNAP29 (12704-1-AP, Proteintech), anti-VAMP3 (10702-1-AP, Proteintech), anti-Phospho-p44/42 MAPK (Erk1/2) (Thr202/Tyr204) (4377, Cell Signaling Technologies), anti-p44/42 Erk1/2 (9107, Cell Signaling Technologies), anti-Phospho-AKT (Ser473) (4060, Cell Signaling Technologies), anti-AKT(pan) (2920, Cell Signaling Technologies).

**RNA Sequencing and Gene Set Analysis**. RNA sequencing libraries were prepared with Truseqv2 RNA library Prep Kit (Illumina) and sequencing on the HiSeq4000 (Illumina). Alignment was performed with STAR and differential gene expression called with Cuffdiff2. Gene set enrichment analysis was run with 1000 permutations, using gene set permutation and weighted Signal2Noise metrics[67–69].

**shRNA and siRNA mediated Gene Knockdown**. Gene knockdowns were performed either with siRNA SMARTpools ordered from Dharmacon or by expressing shRNA from lentivirus produced from a pLKO.1 plasmid system. SiRNAs were introduced to cell using a Lonza nucleofector.

**CRISPR-Cas9 mediated Gene Knockout**. Cas9 was constitutively expressed using lentiviral infection and blasticidin selection. Single guide were delivered with a standard F + E scaffold. Triple guide vectors were similarly expressed with F + E scaffold but with tRNA spacing sequences for individual cleavage[43]. For the tumorigenesis library screen, 500 non-targeting sgRNAs and 500 total targeting sgRNAs against KRAS, HRAS, NRAS, SNAP23, SNAP29, and VAMP3 were constructed. The number of guides chosen was based on the available number of targetable sequences available in the gene exons. The plasmid library was amplified and sequenced on a MiSeq to ensure all guide sequences were represented in the library.

For co-essentiality studies the 1000 guide RNA library were introduced into cell lines stably expressing cas9 using lentiviral infection at a target multiplicity of infection of 0.3. Cells were then selected using puromycin. At the time of cell injection into the subcutaneous space of mice, 1e6 cells were injected into the animal and another 1e6 cells were set aside to control for variances in initial guide RNA representation in the infected pool. Tumors were allowed to grow for four weeks or until they reached the allowed ethical limit for tumor burden per mouse. DNA was extracted using the Qiagen DNA Blood and Tissue kit (69504) and the guide RNA cassettes were amplified and appended to Illumina sequencing arms by PCR. All sequencing was performed with 30% PhiX spike-in controls (Illumina) due to the low complexity of the sequencing library on a MiSeq sequencer. Raw counts were first quantile normalized and the depletion quantitated by fold change from initial cell pool to final tumor. To account for variances in engraftment, the fold change of guides targeting each gene were quantile normalized and queries against specific genes in each tumor were performed in comparison to non-targeting controls.

Due to the large and variable number of guides targeting each gene, and empirical false discovery rate was employed to quantitate the significance of correlations between genes. Each gene-gene correlation is expressed as the mean of all anti-gene1/anti-gene2 guide RNA pairs. For each gene pair, the same number of anti-gene1/anti-gene2 guide RNA correlations were randomly selected from the non-targeting pool were averaged. This calculation was repeated 10,000 times for gene-gene pair, and the percentage of times this random sampling could exceed the actual gene-gene correlation was used as what we refer in the manuscript as the empirical false discovery rate (FDR).

**Cell Proliferation and Anchorage Independent Growth Assays**. Cell proliferation in culture was measured using the CellTiter Blue Cell Viability Assay (Promega) according to manufacturer instructions. Anchorage independent growth assays were done by plating a base layer of 1:1 ratio 1% agar and McCoy's 5A (10%

FBS) then a top layer 1:1 ratio 0.6% agar and McCoy's 5A (10%FBS) with cells seeded at the desired density. Colonies were grown at 37C, 5% $CO_2$ and replenished with media twice per week to avoid dehydration.

**Botulism Toxin E**. For experiments related to botulism toxin isoform E, only the catalytic light chain was used, and the receptor-binding heavily chain was never linked in any experiment in compliance with our established bio-safety level. The in vitro characterization of the catalytic activity profile of both the wild type as well as mutant form of the protease can be found in the manuscript in which it was originally described[54]. The light chain was cloned into the previously described PLEX lentiviral expression construct. Virus production and infection were performed as previously described. Stable expressing cells co-express resistance to puromycin and were therefore selected for under the presence of puromycin (1 mg/ml).

**In vivo Tumorigenesis Studies**. Mouse and human tissue studies were approved by the Stanford institutional review boards and complied with ethical regulations for animal testing and research. Cell lines were first infected with lentiviral vector containing Cas9 and selected in blasticidin for 1 week to obtain a stable population. Cas9-expressing lines were then infected with a lentiviral vector containing single or multiple guide RNA expressing vectors resuspended 2:1 in PBS/matrigel and injected (5e5 for H23 and 5e6 for DLD-1) into the flanks of 8–10 week old. female SHO mice (Crl:SHO-Prkdc$^{scid}$Hr$^{hr}$, Charles River) house in autoclaved cages and fed autoclaved food. These mice were not subject to any other previous experiments. Tumor volumes were measured with caliper measurements taken by two separate investigators and averaged. For CRISPR library screens, cell lines were infected to a target of ×1000 sgRNA coverage and either injected into the subcutaneous space of SHO mice 2 days post-infection or stored at −8 °C for baseline sgRNA representation in the population. Genomic DNA was isolated from cells or tumors using the DNeasy Blood & Tissue Kit (Qiagen). For tissue obtained from tumors, the tissues was disrupted first using manual dissection and homogenization, and the proteinase K digestion step was extended to overnight. All other procedures were done according to the manufacturer's specifications. Guide cassettes were amplified by PCR using PrimeSTAR Max DNA Polymerase Premix (Takara) in two steps (see PCR primers below), and reactions were cleaned up with AMPure XP beads (Beckman Coulter). Library quality control and quantification were done by BioAnalyzer (Agilent Genomics). Libraries were then pooled and sequenced on a MiSeq platform (Illumina) and mapped back to the guide RNA pool. Imperfect matches were discarded and quantification was performed using R.

For all subcutaneous tumor growth experiments, twice the final required number of cells for injection were cultured under the appropriate antibiotic selection for the constructs expressed. At the time of injection, all cell were dissociated from the plate using trypsin, spun down and resuspended into the appropriate volume of culture media. Half of the cells were then injected in a mixture of media and matrigel while the other half were viably frozen for quality control and downstream analysis including immunoblotting, qPCR, and sequencing. For example, in an experiment requiring 1e6 cells injected, greater than 2e6 cells were cultured. 1e6 cells were injected into the mouse and 1e6 were suspended in growth media + 10% DMSO and viably frozen by slow temperature descent in isopropanol. The cell were then revived at a later time and immediately used for downstream experiments.

Library PCR1/2 F:
AATGATACGGCGACCACCGAGATCTACACTCTTTCCCTACACGACGC
TCTTCCGATCTTGTGGAAAGGACGAAACACC
Library PCR1 R:
GTGACTGGAGTTCAGACGTGTGCTCTTCCGATCGTAATACGGTTATCC
ACGCGG
Library PCR2 R:
CAAGCAGAAGACGGCATACGAGATNNNNNNNNNGTGACTGGAGTTCA
GACGTG

**TCGA Patient Survival and Gene Expression Analysis**. Patient data from TCGA including mutations, expression, and clinical outcomes were downloaded through cBioportal (https://www.cbioportal.org/). Statistical tests regarding gene expression were done in cBioportal while survival analysis was done in PRISM.

**Project Achilles Data Analysis**. Raw data was obtained from the Broad institute data portal (https://portals.broadinstitute.org/achilles/datasets/all). The Avana-17Q2-Broad_v2 dataset deposited on 10/30/2017 was used for all analysis in this manuscript. The Pearson correlations for each gene were calculated from all 340 cell lines with complete data available. Pearson's correlation coefficients were measured by the following equation: $p = \Sigma(x_i - x_{mean})(y_i - y_{mean})/[(n-1)s_x s_y]$ from 1 to $n$, where x and y are sample means and s are the respective standard deviations. $n$ is the column length.

Where applicable, the $p$-values for Pearson's correlations are calculated using the t-distribution. Where $t = r$*sqrt(n-2)/sqrt(1-r^2) and $p = 2$*$p(T > t)$ where T is a distribution with $n − 2$ degrees of freedom with relation to $t$.

**Statistics**. Error bars represent standard error of the mean. *$p < 0.05$, **$p < 0.01$, ***$p < 0.001$, ****$p < 0.0001$. Unless otherwise noted, a $t$-test was used to determine statistical significance. In cases where data was not expected to fall in a normal

distribution, Welch's correction was applied. When referring to empirical FDR, *FDR < 0.001.

**Quantitative PCR primers**. SNORD50A: atctcagaagccagatccg, tatctgtgatgatcttatcccgaac
SNORD50B: atctcagaagccgaatccg, taatcaatgatgaaacctatcccgaag
SNORD33: ggccggtgatgagaacttctc gtggcctcagatggtagtgca
SNORD83B: gctggtcagtgatgaggcctg gctggtctcagaaggaaggcaa
Botulism neurotoxin E: cgacaggacaatattgtatattaaacct, gtttttcaagctcgtcggag
L32: aggcattgacaacagggttc, gttgcacatcagcagcactt
KRAS: cccaggtgcgggagaga, tcaaggcactcttgcctacg
The following target sequences for siRNA or shRNA – mediated knockdown:
KRAS: GGAGGGCUUUCUUUGUGUA, UCAAAGACAAAGUGUGUAA, GAAGUUAUGGAAUUCCUUU, GAGAUAACACGAUGCGUAU
KRAS 3'UTR: AUGUUUGGUGUGAAACAAAUUA
HRAS: GAACCCUCCUGAUGAGAGU, AGACGUGCCUGUUUGGACAU, GGAAGCAGGUGGUCAUUGA, GAGGAUGCCUUCUACACGU
NRAS: GAGCAGAUUAAGCGAGUAA, GAAAUACGCCAGUACCGAA, GUGGUGAUGUAACAAGAUA, GCACUGACAAUCCAGCUAA
The following sequences were used for anti-sense oligos:
SNORD50A: CTCAGAAGCCAGATCCGTAA
SNORD50B: CTCAGAAGCCGAATCCGTAC
GFP: TCACCTTCACCCTCTCCACT.

**Reporting Summary**. Further information on research design is available in the Nature Research Reporting Summary linked to this article.

## Data availability

All sequencing data is available from NCBI under the BioProject:PRJNA388817. Processed data including mass spectrometry is available in the supplementary files. Raw files are located in source data. TCGA datasets were downloaded from the cBioPortal website (https://www.cbioportal.org/). Project Achilles data was downloaded from the Broad Institute data portal (https://portals.broadinstitute.org/achilles/datasets/all). All other data and materials are available from the authors upon request.

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

## Acknowledgements

We thank Axel Brunger, Brian Kobilka, Matthew Porteus, Julien Sage, P.J. Utz, Ryan Flynn, and Adam Rubin for valuable discussion in relation to the manuscript. We thank Suzanne Pfeffer for feedback regarding the manuscript and generous usage of the Monolith NT.115. Experiments involving mass spectrometry were performed with the assistance of Chris Adams and Ryan Leib at the Vincent Coates Foundation Mass Spectrometry Laboratory, Stanford University Mass Spectrometry (http://mass-spec. stanford.edu). This work was supported by the USVA Office of Research and Development and by the N.I.H./N.I.A.M.S. grant AR43799 to P.A.K.The project described was also supported, in apart, by Award Numbers 1S10OD01227601 and S10RR027425 from the National Center for Research Resources (NCRR), as well as NIH P30 CA124435 utilizing the Stanford Cancer Institute Proteomics/Mass Spectrometry Shared Resource. Its contents are solely the responsibility of the authors and do not necessarily represent the official views of the NCRR or the National Institutes of Health.

## Author contributions

Y.C. and P.A.K. conceived the project and designed experiments. Y.C., Z.S., J.R.K., T.J., G.W. and L.E. performed the experiments. Y.C. and P.A.K. wrote the manuscript.

## Competing interests

The authors declare no competing interests.
