## [Peer Review File · Nature Communications]

Reviewers' comments:

Reviewer #1 (Remarks to the Author):

In this manuscript, the authors describe a novel direct interaction between K-RAS and the SNARE proteins SNAP23 and SNAP29, as well as implicating components of the late secretory pathway—including SNAP23 and SNAP29—in directing the localization of K-RAS to the plasma membrane. However, while these new findings are exciting, the manuscript at present fails to provide a molecular mechanism to connect their *in vitro* pair-wise protein binding experiments to the cell-level phenotypes they describe. Insight into this mechanism would make this manuscript a stronger candidate for acceptance.

As it stands, the authors do not make any claims about potential mechanisms that would connect SNARE-binding of K-RAS to increased plasma membrane localization. We can imagine two general mechanisms that might be relevant, but unfortunately the cell-based assays performed so far do not allow for these distinct mechanisms to be distinguished.

Mechanism I: K-RAS is incorporated into secretory vesicles, potentially through direct interaction with SNAREs. Disrupting vesicle transport by knock-down of SNAREs globally shuts down movement of cargos to the plasma membrane, including movement of K-RAS.

Mechanism II: K-RAS from the cytoplasm is directly recruited by binding to SNAREs in a vesicle-independent manner. Knock-down of SNAREs that bind to K-RAS may have confounding global blocks in secretion.

Neither of these mechanisms are fully supported by the *in vivo* data presented in Fig. 4. If binding to endosomally-localized SNAP23 and SNAP29 recruits K-RAS to the endosome, as in mechanism II, then, in the TKO strain, there is no reason to expect K-RAS to remain localized to the endosome. However, if K-RAS is incorporated into vesicles in the pathway governed by SNAP23/Vamp3, then, since these two SNAREs act in retrieval from the plasma membrane, these two SNAREs should increase endosomal localization of K-RAS, not decrease it. It is also possible that SNAP23/Vamp3 could indirectly promote movement of K-RAS to the plasma membrane, if they mediate the retrieval to the endosome of some unknown factor that is required for incorporation of K-RAS into anterograde vesicles, but this would not require the direct interaction with K-RAS described earlier in the manuscript.

A first step toward resolving these questions is that, so far, K-RAS localization was only analyzed in a triple knock-down strain, which depletes SNAP23, SNAP29, and Vamp3. This makes it impossible to deconvolute the role of each of these SNAREs individually. However, since these SNAREs act in different pathways, SNAP23/Vamp3 in retrieval and SNAP29 in secretion, single knock outs could have profoundly different effects on K-RAS localization. Further experiments, likely knockdown of additional components of the retrieval and secretion pathways, would be necessary to distinguish between mechanisms I & II.

As a side technical note on the direct binding interactions, the methods are not entirely clear on what specific coding sequences were used. This needs to be made clear. However, if (as seems likely) full-length Vamp3, including the transmembrane helix, was used in the dot-blot in Fig2i, this experiment should probably be repeated with the transmembrane helix removed. Integral membrane proteins, even isolated single-pass helices, are prone to aggregate, and this could decrease the activity of the Vamp3 and prevent the protein analyzed from binding to K-RAS in the experiment, even if a direct interaction does in fact occur.

Reviewer #2 (Remarks to the Author):

Che et al describe the regulation of KRAS by SNORD50. They postulate that SNORD50A/B antagonize the association with KRAS and several SNARE proteins.

The major drawback of the manuscript are a lack of convincing mechanistic insight. Importantly, a role of SNORD50 on polyadenylation has been completely ignored. (Huang C, Shi J, Guo Y, Huang W, Huang S, Ming S, Wu X, Zhang R, Ding J, Zhao W, Jia J, Huang X, Xiang AP, Shi Y, Yao C. A snoRNA modulates mRNA 3' end processing and regulates the expression of a subset of mRNAs. *Nucleic Acids Res.* 2017;45(15):8647-60. doi: 10.1093/nar/gkx651. PubMed PMID: 28911119; PMCID: PMC5587809.)

1. I could not find a characterization of the CRIPSR ko cell lines, there should be northern blots showing the loss of SNORD50A/B expression and a description of the ko strategy, i.e. were the SNORD hosting exons also removed? Also important, was the expression of other SNORDs changed as well.
2. Figure 2A lacks the crucial SCR control. Moreover, a SCR is not a good control, as it is essentially a pool of 4E70 molecules (assuming a length of 70 nt for SNORD50). If there is any sequence selectivity, which is likely, only a subset of these RNAs will bind to KRAS at the given concentration, generating the effect seen in Figure 3B-D. Better controls would be defined snoRNAs or ncRNAs found in the cytosol, like 7SL.
3. The physiological relevance of figure 3 is unclear, as the concentrations are not comparable with cellular concentrations and importantly, the amount of SNORD50 in the cytosol is not quantified. The authors are right that most SNORDs can be detected in the cytosol, but this is only a tiny fraction (see papers from the Ory group, for example Holley CL, Li MW, Scruggs BS, Matkovich SJ, Ory DS, Schaffer JE. Cytosolic accumulation of small nucleolar RNAs (snoRNAs) is dynamically regulated by NADPH oxidase. *The Journal of biological chemistry.* 2015;290(18):11741-8. Epub 2015/03/21. doi: 10.1074/jbc.M115.637413. PubMed PMID: 25792744; PMCID: 4416874.) and look at the levels of SNORDs in non-stimulated cells.
4. In figures 5-6, the different interactions should be rescued by expressing SNORD50A from an expression construct where the SNORD is flanked by exons and there should be no rescue with different SNORDs, if the model is correct.
5. In light of the *Nucleic Acids Res.* 2017;45(15):8647-60 paper, changes in SNARE protein amounts due to SNORD50 knock down should be looked at. The NAR paper lists several hundred changes in polyadenylation and subsequent gene expression due to SNORD50 knockout, which could be compared with data in fig.1 and the supplement.

Reviewer #3 (Remarks to the Author):

In this study Che et al. study the mechanisms whereby the KRAS-binding small RNA SNORD50A/B affect its tumorigenic action. They show that SNORD50A/B RNAs shape KRAS interactome, specifically by inhibiting KRAS proximity to the SNARE vesicular transport proteins SNAP23, SNAP29, and VAMP3. In this respect they demonstrate that SNAREs are essential for KRAS vesicular transport from endosomes to the PM, with KRAS localization to the PM facilitated by SNAREs but antagonized by SNORD50A/B. In this context, depletion of SNARE proteins markedly compromises KRAS signals and tumorigenesis. Accordingly, cancer data analyses indicate a functional KRAS dependency on these SNAREs. This is indeed an interesting piece of work, of unquestionable novelty, that offers a new perspective on KRAS trafficking and its implications in KRAS-driven tumorigenesis. However, some

controls on several aspects that are not fully clear and some additional experiments on some points pertinent to the main tenet of the study, would be required to make the story fully convincing.

1) Figure 2. ... "The prior work on KRAS and SNORD50A/B identified KRAS Lys5, Lys42, Arg149 and Arg161 residues as necessary for SNORD50A/B binding. Mutagenesis of these same residues disrupted SNARE interactions without altering interaction with Raf1 (Figure 2C,D)".... This is not shown in Fig 2C,D which refers to the deletion of the switch and hypervariable regions. Evidently, this is a mishap that should be amended. This piece of data is important.

2) Since Lys5, Lys42, Arg149 and Arg161 are conserved among H, K and NRAS: why do SNAREs specifically bind to KRAS and not to H or NRAS?. Can purified SNAREs bind H or NRAS in a cell-independent context? (i.e. in vitro?)

3) Authors mention that Fig S2G reveals a preferential proximity of SNAP23 and 29 to KRAS. I cannot agree with this statement. Whereas SNAP23 is clear, SNAP29 proximity to H and KRAS is identical. In this respect, PLAs / co-IPs should be performed also for HRAS / NRAS vs the three SNARE proteins.

4) Since SNAREs specificity for KRAS cannot be explained based on some specific binding motif on KRAS, then could it be that SNAREs colocalize with KRAS somewhere in the cell where N and HRAS are absent?. Subcellular localization is a major factor in this study, so to study the colocalization of the SNAREs; the three RAS isoforms and relevant plasma-membrane and endomembrane markers is important. Fig 4 B-D demonstrate that KRAS, but not H or NRAS relocalize in the absence of SNAREs, but this does not tell us exactly where SNAREs and RAS co-localize, and alternative interpretations to that given by authors could explain such relocalization.

5) Does overexpression of SNORD50 result in KRAS accumulation in endosomes?

6) PDEdelta has been shown to play a major role on KRAS trafficking between plasma-membrane and endomembranes: how do SNORD50 and SNAREs affect KRAS binding to PDEdelta?

7) How does SNAREs depletion affect activation of wt KRAS by GEFs?. Arguably, if SNAREs depletion locks KRAS in endosomes: will it be available for activation by, for example, SOS or RASGRF?.

8) Authors demonstrate SNAREs depletion affects KRAS signal output in cells harboring oncogenic KRAS, such as DLD-1. Would overexpression of SNORD50 in this cells have similar effects?

Response to Reviewers: KRAS regulation by small non-coding RNAs and SNARE proteins

Authors: Che, Y., Siprashvili, Z., Kovalski, J.R., Jiang T., Wozniak G., Khavari, P.A.*

1 Reviewer 1

1.1 In this manuscript, the authors describe a novel direct interaction between K-RAS and the SNARE proteins SNAP23 and SNAP29, as well as implicating components of the late secretory pathway including SNAP23 and SNAP29 in directing the localization of K-RAS to the plasma membrane. However, while these new findings are exciting, the manuscript at present fails to provide a molecular mechanism to connect their in vitro pair-wise protein binding experiments to the cell-level phenotypes they describe. Insight into this mechanism would make this manuscript a stronger candidate for acceptance.

We thank the Reviewer for the time and careful consideration of the manuscript as well as the kind words regarding the novelty of the proposed work. We agree that greater insight into the mechanism would strengthen the manuscript and we have undertaken extensive additional experimentation and revision guided by the Reviewer's point-by-point comments in order to improve the manuscript.

1.2 As it stands, the authors do not make any claims about potential mechanisms that would connect SNARE-binding of K-RAS to increased plasma membrane localization. We can imagine two general mechanisms that might be relevant, but unfortunately the cell-based assays performed so far do not allow for these distinct mechanisms to be distinguished.

Mechanism I: K-RAS is incorporated into secretory vesicles, potentially through direct interaction with SNAREs. Disrupting vesicle transport by knock-down of SNAREs globally shuts down movement of cargos to the plasma membrane, including movement of K-RAS.

Mechanism II: K-RAS from the cytoplasm is directly recruited by binding to SNAREs in a vesicle-independent manner. Knock-down of SNAREs that bind to K-RAS may have confounding global blocks in secretion.

Neither of these mechanisms are fully supported by the in vivo data presented in Fig. 4. If binding to endosomally-localized SNAP23 and SNAP29 recruits K-RAS to the endosome, as in mechanism II, then, in the TKO strain, there is no reason to expect K-RAS to remain localized to the endosome. However, if K-RAS is incorporated into vesicles in the pathway governed by SNAP23/Vamp3, then, since these two SNAREs act in retrieval from the plasma membrane, these two SNAREs should increase endosomal localization of K-RAS, not decrease it. It is also possible that SNAP23/Vamp3 could indirectly promote movement of K-RAS to the plasma membrane, if they mediate the retrieval to the endosome of some unknown factor that is required for incorporation of K-RAS into anterograde vesicles, but this would not require the direct interaction with K-RAS described earlier in the manuscript.

We thank the Reviewer for the thoughtful critiques and agree that further experiments including those suggested below would further elucidate the mechanistic relationship between SNARE interactions with KRAS. Both Mechanism I and Mechanism II are excellent hypotheses and we have integrated them into our follow up steps. We outline the experiments undertaken in conjunction with the Reviewer's suggested experiments below.

1.3 A first step toward resolving these questions is that, so far, K-RAS localization was only analyzed in a triple knock-down strain, which depletes SNAP23, SNAP29, and Vamp3. This makes it impossible to deconvolute the role of each of these SNAREs individually. However, since these SNAREs act in different pathways, SNAP23/Vamp3 in retrieval and SNAP29 in secretion, single knock outs could have profoundly different effects on K-RAS localization. Further experiments, likely knockdown of additional components of the retrieval and secretion pathways, would be necessary to distinguish between mechanisms I II.

We agree with the Reviewer's suggestion to perform KRAS localization experiments in individual SNARE KO in order to further clarify which individual proteins are the most individually consequential in the proposed mechanism of sub-cellular localization.

In order to answer this question, we individually performed SNAP23, SNAP29, and VAMP3 KOs alongside Non-targeting gRNA controls using CRISPR-cas9 as previously described (**New Figure S7A**). We confirmed comparable levels of KO as prior and split the remaining pool of cells into both IF and subcellular fractionation assays (**New Figure S7B**). We find that:

- 1) SNAP23 KO most closely phenocopies the relocalization phenotype of the triple KO, resulting in KRAS localization to Arf6-labeled recycling endosomes (**New Figure S7A**).
- 2) SNAP29 and VAMP3 KO alone do not specifically relocalize KRAS to recycling endosomes (**New Figure S7A**).
- 3) Neither SNAP23, SNAP29, nor VAMP3 cause a significant shift between total membrane and total cytoplasmic distribution of KRAS (**New Figure S7B,C**).

We therefore concluded that SNAP23 is the primary driver of the described phenotype of mislocalization and the described phenotypes. This is additionally supported by our previous observations that SNAP23 knockout is significantly correlated with KRAS essentiality in both our multi-cell line in vivo tumorigenesis co-essentiality studies (**Figure 7A,B**) as well as the larger co-essentiality database in the Achilles library of cell-culture dependencies (**Figure S6G**). Originally we also observed that SNAP23 was the most selective for KRAS proximity interaction by BioID. We further confirmed that this finding was true for direct in vitro interaction by performing Far western experiments where SNAP23, SNAP29, and VAMP3 were fixed to a nitrocellulose membrane, KRAS/HRAS/NRAS were allowed to bind from solution, and then detected via Ras-specific antibodies (**Figure 2I,J**; **New Figure S3A**).

Using these studies as a guide, we focused on further studying the SNAP23-KRAS interaction and how it might drive localization. As the Reviewer describes, SNAP23 is well-implicated in mediating the plasma membrane side of vesicle fusion with the PM as well as retrieval. Neither of these steps, however, drive transport from recycling endosomes to the PM through direct interaction. It has also been well-described, however, that SNAP23 has a robust presence in recycling endosomes (**PMID:9817754**, **PMID:9647644**, **PMID:9651373**). We hypothesized that direct interaction of KRAS with SNAP23 on recycling endosomes themselves may allow KRAS to obtain a more favorable spatial positioning on recycling endosomes favorable for PM transport, effectively piggy-backing the PM transport mechanism of SNAP23.

In order to test this, we undertook super-resolution microscopy to obtain high resolution images of the spatial relationship of KRAS and SNAP23 in recycling endosomes (**New Figure S8A**). In order to overcome cross-reactivity of the secondary antibodies, we prelabelled KRAS, SNAP23, and the recycling endosome marker Arf6 with fluorophores in separate excitation/emission ranges and used them to label A549s.

As expected, we find close proximal relationships between SNAP23 and KRAS in areas dense with Arf6 signal (**New Figure S8A,B**). We then overexpressed SNORD50 in the same cells and performed the same imaging experiments. In the context of SNORD50 overexpression, we expect competitive inhibition of the SNAP23-KRAS binding interaction as previously demonstrated (**Figure 3B,C**). Under super-resolution imaging, we see greater spatial separation of KRAS from SNAP23 (**Figure S8A,B**) and additionally, the phenocopy of SNAP23 KO in global relocalization (**New Figure S7A**). These experiments appear to support a mechanism by which spatial proximity to SNAP23 on recycling endosome enables PM transport out of recycling endosomes.

1.4 As a side technical note on the direct binding interactions, the methods are not entirely clear on what specific coding sequences were used. This needs to be made clear. However, if (as seems likely) full-length Vamp3, including the transmembrane helix, was used in the dot-blot in Fig2i, this experiment should probably be repeated with the transmembrane helix removed. Integral membrane proteins, even isolated single-pass helices, are prone to aggregate, and this could decrease the activity of the Vamp3 and prevent the protein analyzed from binding to K-RAS in the experiment, even if a direct interaction does in fact occur.

We thank the Reviewer for the detailed technical critique and agree in general with the assessment of the above considerations. The Reviewer is correct that full length recombinant VAMP3 was used in our direct interaction studies, and this is now stated clearly in the revised manuscript.

We agree that aggregation of highly hydrophobic segments of proteins physiologically used for membrane intergration can be a source of protein aggregation and therefore confound the results of interaction studies in vitro. We however chose to use full length VAMP3 given existing evidence that full length VAMP3 retains its ability to bind its known partners like the Human Herpesvirus 6 gM/gN complex when reconstituted in vitro (PMID: 25209806), suggesting that the degree of aggregation that may occur is not itself sufficient to cause a false negative result.

2 Reviewer 2

2.1 Che et al describe the regulation of KRAS by SNORD50. They postulate that SNORD50A/B antagonize the association with KRAS and several SNARE proteins.

The major drawback of the manuscript are a lack of convincing mechanistic insight. Importantly, a role of SNORD50 on polyadenylation has been completely ignored. (Huang C, Shi J, Guo Y, Huang W, Huang S, Ming S, Wu X, Zhang R, Ding J, Zhao W, Jia J, Huang X, Xiang AP, Shi Y, Yao C. A snoRNA modulates mRNA 3' end processing and regulates the expression of a subset of mRNAs. *Nucleic Acids Res.* 2017;45(15):8647-60. doi: 10.1093/nar/gkx651. PubMed PMID: 28911119; PMCID: PMC5587809.)

We thank the Reviewer for the thoughtful comments and constructive feedback. Through further experimentation and clarification of existing data, we believe we have addressed the concerns as well as further clarifying the proposed mechanism of SNARE/SNORD50 interactions with KRAS. We elaborate on the details in a point-by-point manner below.

2.2 I could not find a characterization of the CRISPR ko cell lines, there should be northern blots showing the loss of SNORD50A/B expression and a description of the ko strategy, i.e. were the SNORD hosting exons also removed? Also important, was the expression of other SNORDs changed as well.

We apologize that these data were not more clearly referred to in the manuscript. These CRISPR KO cell lines were generated as a part of the previously described work (Siprashvili et al. 2016). The lines were generated with a two-cut strategy where the snoRNAs were excised completely from the genomic sequence in both chromosomes without disrupting the host gene. The clonal disruption was verified by DNA blots verifying a single PCR band of the expected shortened genomic length flanking the excised region as well as Sanger sequencing (Siprashvili et al. 2016, Figure 3H, Figure S5A,D,G,J,M). As this strategy targeted a specific genomic locus and did not use any of the shared snoRNA homology sequences as a mechanism of knockdown, there should be no change in expression of other snoRNAs. We agree, however, that it is important that no global effect on canonical snoRNA modulation of ribosomal modification and function is affected by SNORD50A/B KO. Thus, ribosomal profiling was performed in our previous work (Siprashvili et al. 2016, Figure S5B,C). SNORD50A/B KO resulted in no global changes in ribosome occupancy or pausing index when compared to ribosome stress-inducing agents amino acid analog Lazetidine-2-carboxylic-acid and the proteasome inhibitor MG132, thus indicating no global dysfunction in canonical snoRNA biology. We have revised the manuscript to more clearly reference this previous data and apologize for the ambiguity in the original draft.

2.3 Figure 2A lacks the crucial SCR control. Moreover, a SCR is not a good control, as it is essentially a pool of 4E70 molecules (assuming a length of 70 nt for SNORD50). If there is any sequence selectivity, which is likely, only a subset of these RNAs will bind to KRAS at the given concentration, generating the effect seen in Figure 3B-D. Better controls would be defined snoRNAs or ncRNAs found in the cytosol, ile 7SL.

We thank the Reviewer for the excellent recommendation regarding controls for our RNA interactions with KRAS. Accordingly, we have performed the SCR RNA control on MST, finding no detectable interaction in the queried concentration range in biological triplicate with recombinant KRAS under identical conditions to previously used (New Figure 3A). We additionally agree that the specificity of interaction should be verified against other RNAs including both other snoRNAs as well as cytoplasmic transcripts of other RNA classes. In our previous work, we describe KRAS interaction strengths with the following RNAs: SNORD50A, SNORD50B, SNORD53, SNORD123, SNORD124, SNORD126, SNORD33, SNORD76, SNORA45, SNORA12, SNORA22, SNORA24, BCL2sense, BCL2antisense, MYCsense, MYCantisense, HRASsense, HRASantisense, TP53sense, TP53antisense, DLEU1sense, DLEU1antisense, HOTAIRsense, HOTAIRantisense, IGF2RNCsense, IGF2RNCantisense, OCC1sense, OCC1antisense, PWRN1sense, PWRN1antisense, SOX2OTsense, and SOX2OTantisense. KRAS demonstrated high binding specificity for SNORD50A/B interaction among all of the tested RNAs (Siprashvili et al. 2016, Supplementary Table 2).

2.4 The physiological relevance of figure 3 is unclear, as the concentrations are not comparable with cellular concentrations and importantly, the amount of SNORD50 in the cytosol is not quantified. The authors are right that most SNORDs can be detected in the cytosol, but this is only a tiny fraction (see papers from the Ory group, for example Holley CL, Li MW, Scruggs BS, Matkovich SJ, Ory DS, Schaffer JE. Cytosolic accumulation of small nucleolar RNAs (snoRNAs) is dynamically regulated by NADPH oxidase. The Journal of biological chemistry. 2015;290(18):11741-8. Epub 2015/03/21. doi: 10.1074/jbc.M115.637413. PubMed PMID: 25792744; PMCID: 4416874.) and look at the levels of SNORDs in non-stimulated cells.

We thank the Reviewer for raising this point and giving us the opportunity to better clarify the purpose of Figure 3 and supplement additional experiments to clarify the mechanism by which we believe SNORD50 and SNAREs regulate KRAS localization.

Our hypothesis is that SNORD50 competitively inhibits KRAS interactions with SNAP23 based on previous and new experiments summarized below: 1) Microscale thermophoresis (**Figure 3A,C,D**) and far western blotting (**Figure 2I, Figure 3B**) demonstrating direct interaction of KRAS with SNAP23 that is then inhibited by SNORD50. 2) Specificity of the KRAS-SNAP23 interaction among Ras isoforms by far western blotting (**New Figure S3A**).

We further refined our previous subcellular localization assays with the following experiments: 1) Individual knock out of SNAP23, SNAP29, and VAMP3, identifying SNAP23 as the crucial interactor in recycling endosome - to plasma membrane transport (**New Figure S7A**). 2) SNORD50 overexpression resulting in partial phenocopy of SNAP23 KO, supporting competitive interaction with KRAS thus leading to impaired PM transport (**New Figure S8A**).

In order for the proposed mechanism to be correct, the interaction of KRAS with SNORD50 should predominantly be at recycling endosomes, a membrane component. This was the original intention of Figure 3. We agree with the Reviewer that this could be stated more clearly and have thus clarified the text to better outline our reasoning. We also agree that this mechanism could be more thoroughly explored by additional experimentation.

Accordingly, we undertook more detailed super-resolution microscopy with co-staining of KRAS, SNAP23, and Arf6 in order to track the relative spatial localization of KRAS with SNAP23 in the Arf6-labeled recycling endosome compartment. We find the following: 1) Close spatial proximity between KRAS and SNAP23 in Arf6-labeled recycling endosomes in A549s (**New Figure S8B**), the cell line in which we undertook the majority of our other localization experiments. In these same WT conditions, we observe the expected PM enrichment of KRAS (**Figure S4B**). 2) SNORD50 overexpression in these same cells results in greater spatial distance between KRAS and SNAP23 in recycling endosomes as well as increasing the amount of KRAS trapped in the recycling endosome compartment (**New Figure S8B,C**).

These experiments together support a model where KRAS binds to SNAP23 in order to enrich its localization onto a membrane subcompartment of the recycling endosome destined for the plasma membrane. This interaction is competitively inhibited by SNORD50 binding to KRAS, which leads to entrapment in the recycling endosomes (**New Figure S8A**) where KRAS has a diminished ability to effectively relay RTK signaling to the cytoplasmic kinase cascades downstream (**Figure S5A-F**) and diminishes its oncogenic potential (**Figure 6A-G**).

2.5 In figures 5-6, the different interactions should be rescued by expressing SNORD50A from an expression construct where the SNORD is flanked by exons and there should be no rescue with different SNORDs, if the model is correct.

We thank the Reviewer for pointing out that this portion of the manuscript was not clear. In our proposed mechanism SNORD50 expression should result in decrease interaction between KRAS with SNAP23 and therefore decrease KRAS downstream signaling. We confirm this by overexpressing SNORD50 in a transfection vector with flanking exons which partially phenocopies SNAP23 KO (**New Figure S7A, New Figure S8A**). In addition, we confirmed the specificity of KRAS interaction with specifically SNORD50A/b in the above mentioned interaction studies between KRAS and SNORD50A, SNORD50B, SNORD53, SNORD123, SNORD124, SNORD126, SNORD33, SNORD76, SNORA45, SNORA12, SNORA22, SNORA24, BCL2sense, BCL2antisense, MYCsense, MYCantisense, HRASsense, HRASantisense, TP53sense, TP53antisense, DLEU1sense, DLEU1antisense, HOTAIRsense, HOTAIRantisense, IGF2RNCsense, IGF2RNCantisense, OCC1sense, OCC1antisense, PWRN1sense, PWRN1antisense, SOX2OTsense, and SOX2OTantisense (**Siprashvili et al. 2016, Supplementary Table 2**).

2.6 In light of the Nucleic Acids Res. 2017;45(15):8647-60 paper, changes in SNARE protein amounts due to SNORD50 knock down should be looked at. The NAR paper lists several hundred changes in polyadenylation and subsequent gene expression due to SNORD50 knockout, which could be compared with data in fig.1 and the supplement.

We thank the Reviewer for this suggestion. Although the experiments were performed in different cell types, one important null hypothesis of our initial proximity proteomics experiment is that SNORD50 KO results in increased stability of the SNARE proteins mRNA transcripts and therefore increased protein expression, leading us to the observation of increased KRAS proximity.

In our analysis of the data from *Nucleic Acids Res.* 2017;45(15):8647-60 we find that SNAP23, SNAP29, and VAMP3 are not upregulated in response to SNORD50 KD. In fact, none of the proteins that increase their proximal interaction with KRAS are upregulated in the above study (**New Figure 2H**). Therefore the mechanism by which SNORD50 alters KRAS interaction with SNAREs is unlikely to be an indirect mechanism through its functions on polyadenylation and subsequent gene expression. We now comment on this comparison directly in the manuscript.

3 Reviewer 3

3.1 In this study Che et al. study the mechanisms whereby the KRAS-binding small RNA SNORD50A/B affect its tumorigenic action. They show that SNORD50A/B RNAs shape KRAS interactome, specifically by inhibiting KRASproximity to the SNARE vesicular transport proteins SNAP23, SNAP29, and VAMP3. In this respect they demonstrate that SNAREs are essential for KRAS vesicular transport from endosomes to the PM, with KRAS localization to the PM facilitated by SNAREs but antagonized by SNORD50A/B. In this context, depletion of SNARE proteins markedly compromises KRAS signals and tumorigenesis. Accordingly, cancer data analyses indicate a functional KRAS dependency on these SNAREs. This is indeed an interesting piece of work, of unquestionable novelty, that offers a new perspective on KRAS trafficking and its implications in KRAS-driven tumorigenesis. However, some controls on several aspects that are not fully clear and some additional experiments on some points pertinent to the main tenet of the study, would be required to make the story fully convincing.

We thank the Reviewer for the kind words and the constructive feedback. In order to further elucidate the functional link between KRAS trafficking and SNAREs, we undertook the suggested experiments and revisions. We outline the results in a point-by-point manner below.

3.2 Figure 2. ...”The prior work on KRAS and SNORD50A/B identified KRAS Lys5, Lys42, Arg149 and Arg161 residues as necessary for SNORD50A/B binding. Mutagenesis of these same residues disrupted SNARE interactions without altering interaction with Raf1 (Figure 2C,D)”.... This is not shown in Fig 2C,D which refers to the deletion of the switch and hypervariable regions. Evidently, this is a mishap that should be amended. This piece of data is important.

We apologize for the confusion in Figure 2C,D. The suggested KRAS (Lys5, Lys42, Arg149 and Arg161) is in fact included in that experiment but was labeled the "RNA" mutant due to space and formatting constraints of the figure. We clarify this labeling shorthand in the body of the manuscript when referring to **Figure 2C,D**.

3.3 Since Lys5, Lys42, Arg149 and Arg161 are conserved among H, K and NRAS: why do SNAREs specifically bind to KRAS and not to H or NRAS?. Can purified SNAREs bind H or NRAS in a cell-independent context? (i.e. in vitro?)

The reviewer makes an excellent point that Lys5, Lys42, Arg149 and Arg16 are conserved residues between the three major Ras isoforms. It therefore raises the question which of the observed SNARE interactions if any are common to all Ras proteins and which are more specific to KRAS.

In order to answer this question, we undertook qualitative far western binding experiments under the same technical parameters are previously described in the manuscript (**Figure 2I, New Figure S3A**). Previously, we had observed that SNAP23 and SNAP29 have the capacity to bind KRAS in a cell-independent fashion. We repeated the far western experiments for HRAS and NRAS - SNAP23, SNAP29, and VAMP3 were spotted in serial dilution onto a nitrocellulose membrane onto which recombinant HRAS or NRAS was flowed and allowed to come to equilibrium. The excess protein in solution was then washed away, and SNARE-bound HRAS or NRAS detected by a pan-ras antibody by western blot. We find that:

- 1) As previously described for KRAS, HRAS and NRAS also do not directly bind VAMP3.
- 2) SNAP29 is capable of binding all three Ras isoforms.
- 3) SNAP23 is a KRAS-specific interactor that does not bind HRAS or NRAS in vitro.

These results are consistent with the trends seen in our proximity proteomics experiments comparing the vicinal proteome of the

three Ras isoforms (**Figure S2G**) where we also note that SNAP23 is the most specific to KRAS. While it is true that the particular residues that are important for SNARE interaction on KRAS are conserved on HRAS and NRAS, it has been well established that the high sequence homology leads to distinct biochemical and allosteric differences between the proteins (**PMID: 28630043**, **PMID: 30194290**), and in this case leading to a difference in interaction with SNAP23.

3.4 Authors mention that Fig S2G reveals a preferential proximity of SNAP23 and 29 to KRAS. I cannot agree with this statement. Whereas SNAP23 is clear, SNAP29 proximity to H and KRAS is identical. In this respect, PLAs / co-IPs should be performed also for HRAS / NRAS vs the three SNARE proteins.

We thank the Reviewer for raising this point and agree that the specificity for SNARE-KRAS interaction is strongest for SNAP23. After reviewing the original data, we agree with the Reviewer's point and have revised the text of the manuscript to reflect as such. We apologize for the original misinterpretation and confusion. We additionally performed far western blotting with recombinant HRAS and NRAS to confirm the selectivity of the KRAS-SNAP23 interaction and the lack of isoform specificity between the Ras-SNAP29 (**New Figure S3A**). As mentioned above, SNAP23 has high isoform selectivity in direct binding for KRAS, but SNAP29 appears to bind NRAS, HRAS, and KRAS. VAMP3 does not appear to directly bind any of the Ras isoforms.

3.5 Since SNAREs specificity for KRAS cannot be explained based on some specific binding motif on KRAS, then could it be that SNAREs colocalize with KRAS somewhere in the cell where N and HRAS are absent?. Subcellular localization is a major factor in this study, so to study the colocalization of the SNAREs; the three RAS isoforms and relevant plasma-membrane and endomembrane markers is important. Fig 4 B-D demonstrate that KRAS, but not H or NRAS relocalize in the absence of SNAREs, but this does not tell us exactly where SNAREs and RAS co-localize, and alternative interpretations to that given by authors could explain such relocalization.

While it is true that KRAS shares high homology with the other Ras isoforms at the sites of proposed binding, we demonstrate above that there is significant selectivity of the SNAP23 interaction with KRAS that is true both in vivo and in vitro (**Figure 2I,J**, **New Figure S3A**, **Figure S2G**). It is therefore likely that the interaction itself is specific and not simply dependent on subcellular co-localization.

In order to answer the question of whether or not the co-localization is significant, we followed up our previous ultra-resolution microscopy with costaining of KRAS, SNAP23, and Arf6 in the WT and SNORD50 overexpression in order to study the effects of inhibiting SNAP23 interaction with KRAS on KRAS localization (**New Figure S8B,C**). In the WT condition, we find that as previously reported, we see close spatial proximity of SNAP23 with KRAS in area co-stained with Arf6, suggesting recycling endosome interaction. In the context of SNORD50 overexpression, we find that this spatial proximity is disrupted, with KRAS and SNAP23 localizing further apart. In the same conditions of SNORD50 o/e, we also note the entrapment of KRAS into recycling endosomes, supporting the functional consequence of this spatial separation (**New Figure S8A**). We propose that the KRAS uses SNAP23 to find the subsection of membrane in recycling endosome that are actively being transported to the plasma membrane.

3.6 Does overexpression of SNORD50 result in KRAS accumulation in endosomes?

This is an excellent question, and we have undertaken experiments to directly address this possibility.

SNORD50A/B sequences were cloned into transient expression vectors with flanking sequences identical to the endogenous exonic sequences. These vectors were then expressed in A549s with expression confirmed by quantitative PCR (**New Figure S4F**). Additionally, the delivery vector was supplanted with a constitutive expressing DSred marker so that active expression could also be confirmed by imaging.

Imaging performed identically to previously described reveal the same wide-spread distributed as found in prior experiments in the WT/mock infection condition. The overexpression of SNORD50, however, creates a visible focus of KRAS signal in the recycling endosome distribution, confirmed by co-stain with Arf6 (**New Figure S8A**). This is similar to the change in subcellular distribution observed with SNAP23 KO (**New Figure S7A**), but notably without as striking a depletion of KRAS in other cellular compartments. This leads us to hypothesize that SNORD50 overexpression does in fact inhibit SNARE-mediated transport out of the recycling endosome, but that this process can be competitively overcome, consistent with the proposed mechanism of action.

3.7 PDEdelta has been shown to play a major role on KRAS trafficking between plasma-membrane and endomembranes: how do SNORD50 and SNAREs affect KRAS binding to PDEdelta?

We thank the Reviewer for this observation, and as the Reviewer correctly points out, PDEd has been found to regulated the subcellular transport of KRAS by facilitating the cytoplasmic solubility of the C-terminus. To answer the question of whether SNARE disruption or SNORD50 expression change the interaction of KRAS and PDEd in cells, we performed PLA with paired antibodies targeting RAS and PDEd respectively in a KRAS-dominant line. The interaction rates of KRAS with PDEd does not significantly change in the context of SNARE KO or SNORD50A/B overexpression (**New Figure S4D,E**), suggesting that SNAP23/SNORD50A/B-mediated transport is independent of PDEd.

Among the different conditions, there is no statistically significant difference in frequency of interaction between KRAS and PDEd, which suggests that the stage of transport mediated by the SNARE/SNORD50 axis is in large part independent of PDEd action on KRAS.

3.8 How does SNAREs depletion affect activation of wt KRAS by GEFs?. Arguably, if SNAREs depletion locks KRAS in endosomes: will it be available for activation by, for example, SOS or RASGRF?.

This is a fascinating question that may extend beyond the scope of this study and certainly deserves further investigation. It is well known that the Ras GTPases have some basal function within endosomal compartments but there is considerable variation among isoforms of Ras (**PMID: 26921695**).

In order to directly answer the Reviewer's question, we expressed HA-tagged WT KRAS in A549 and performed PLA experiments detecting the proximity of antibodies specific to the HA-tag and antibodies specific to GTP-bound Ras. This allowed us to detect the rate of Ras activation by GEFs indirectly through the relative amounts of GTP-bound WT KRAS. This experiments revealed that complete serum depletion, as expected, resulted in a decrease in GTP-bound Ras, but interestingly there was no significant change with SNARE or SNORD50A/B KO suggesting that there is enough basal GDP to GTP conversion as to not be the rate limiting step by which SNAREs and SNORD50A/B disrupt KRAS (**New Figure S4I,J**). Instead the critical changes may instead be downstream in the effector kinase cascades, which is supported by the differential changes in interaction frequency in KRAS with PI3K and RAF1 signaling pathways (**Figure 5A-D**) and also consistent with previous data suggesting that KRAS may have crucial differences in its preferential activation of certain downstream pathways both on the PM as well as in endosomes (**PMID: 12077341, PMID: 19289794**).

3.9 Authors demonstrate SNAREs depletion affects KRAS signal output in cells harboring oncogenic KRAS, such as DLD-1. Would overexpression of SNORD50 in this cells have similar effects?

We thank the reviewer for this question and outline the experiments undertaken to answer it below. SNORD50 was overexpressed in the DLD-1 KRAS WT/G12D as previously described. The cells were then harvested and protein lysate run on western blot. Total ERK and AKT with phosphorylated ERK and AKT were blotted and quantified in biological triplicate. In contrast to in vivo tumorigenesis, cultured DLD-1 appeared to have minimal activation of the AKT signaling axis compared to ERK. With SNORD50A/B o/e, there was little detectable change in pAKT levels, but significant decrease in pEKT levels, suggesting that SNORD50 o/e does disrupt the ERK signaling axis downstream of KRAS (**New Figure S6E**).

Reviewers' comments:

Reviewer #1 (Remarks to the Author):

The authors deserve to be commended on the thoroughness with which they have responded to all reviewer comments and suggestions. The revised manuscript is much improved, incorporates a substantial quantity of new data, and should in my opinion be published promptly.

Reviewer #2 (Remarks to the Author):

Despite new experiments and a tremendous amount of data, key questions related to the SNORD-aspect are still open: Does SNORD50A/B bind in vivo to KRAS and are the observed effects due to specific directed binding to this SNORD?

Figure 3A does not contain any SCR control data, as stated in the rebuttal letter 2.3, neither in the figure nor figure legend to 3A

In response to the physiological relevance, additional experiments were performed, but the key experiment: is SNORD50 localized in endosomes is not addressed, which is puzzling. Can the authors detect SNORD50 in the cytosol and in purified endosomes? How does this compare to the concentration found in the nucleus/nucleolus?

Figure S8B, C show colocalization of proteins KRAS/SNAP23 and if I interpret the data correctly, the analysis is based on six and seven cells (right panel). This is way too indirect to support physiological binding of SNORD50. The same holds true for S4B.

In response to 2.5, a specificity control, namely the expression of another SNORD is missing (SNORD expression construct flanked by two non-coding exons, a SNORD different from SNORD50)

Reviewer #3 (Remarks to the Author):

The authors have addressed the raised concerns to my satisfaction

Response to Reviewers: KRAS regulation by small non-coding RNAs and SNARE proteins

Authors: Che, Y., Siprashvili, Z., Kovalski, J.R., Jiang T., Wozniak G., Elcavage L, Khavari, P.A.*

Verbatim comments in italics, responsive revisions in blue.

Reviewer #1 (Remarks to the Author):

“The authors deserve to be commended on the thoroughness with which they have responded to all reviewer comments and suggestions. The revised manuscript is much improved, incorporates a substantial quantity of new data, and should in my opinion be published promptly.”

We thank the Reviewer for the thoughtful input and are grateful to have had the opportunity to revise and improve the manuscript guided by their excellent suggestions in the prior rounds of review.

Reviewer #2 (Remarks to the Author):

“Despite new experiments and a tremendous amount of data, key questions related to the SNORD-aspect are still open: Does SNORD50A/B bind in vivo to KRAS and are the observed effects due to specific directed binding to this SNORD?”

We thank the Reviewer for suggestions in the first review and again appreciate the thoughtful comments regarding the revised manuscript. Guided by the questions, we have now added the following additional new data below to the manuscript and address each question further below in a point-by-point manner.

- 1)** Direct visualization of SNORD50A/B co-localization with KRAS in recycling endosomes by super-resolution microscopy of SNORD50A/B-targeting antisense oligos (**New Supplemental Figure 8B**).
- 2)** Protein microarray binding of 9,125 recombinant proteins spotted in duplicate for the C/D snoRNAs SNORD33 and SNORD83B performed under identical conditions to that previously reported for SNORD50A/B (**New Supplemental Data Table 8, 9, 10, and 11**). SNORD33 and SNORD83B are now included in new in vivo controls as suggested.
- 3)** Super resolution microscopy of KRAS, SNAP23, and ARF6 as previously described now with added SNORD33 and SNORD83B overexpression controls as previously performed for SNORD50A/B (**New Supplemental Figures 8C, D, and E**) demonstrating the specificity of SNORD50A/B in displacing KRAS-SNAP23 interaction.

With regard to the evidence of direct binding between KRAS and SNORD50A/B in vivo, we previously performed UVC crosslinking of live cells, which created amino-acid/RNA nucleotide crosslinks specifically at direct-angstrom interactions (**PMID:**

30068528). These experiments demonstrated direct in vivo binding interactions between KRAS and SNORD50A/B and not other related snoRNAs SNORD33, SNORD83B, SNORD86, SNORD92, SNORA3, and SNORA67 (**Siprashvilli et al. 2016 Figure 2D**).

To address the question of specificity, we directly detect SNORD50A/B in recycling endosomes by super-resolution microscopy (**New Supplemental Figure 8B**) and add additional controls to demonstrate that SNORD33 and SNORD83B do not have significant in vitro binding capacity for KRAS (**New Supplemental Data Table 8, 9, 10, and 11**) and that their overexpression in cells does not displace KRAS-SNAP23 interactions whereas SNORD50A/B overexpression does (**New Supplemental Figures 8C, D, and E**).

“Figure 3A does not contain any SCR control data, as stated in the rebuttal letter 2.3, neither in the figure nor figure legend to 3A”

We apologize for the confusion with regard to this figure. SCR RNA binding by microscale thermophoresis was performed in biological triplicate with recombinant KRAS with no detectable binding. We denote this with the legend in the top left corner of the figure subsection (**Figure 3A**). When there is insufficient fractional binding difference in the concentration range tested, however, no dissociation curve can be generated since there is no detectable binding interaction to change the fluorescent particle movement. We clarify this further in the materials and methods of the manuscript.

“In response to the physiological relevance, additional experiments were performed, but the key experiment: is SNORD50 localized in endosomes is not addressed, which is puzzling. Can the authors detect SNORD50 in the cytosol and in purified endosomes? How does this compare to the concentration found in the nucleus/nucleolus?”

We thank the Reviewer for further emphasizing this point and bringing it up as an opportunity to further investigate the biology of SNORD50A/B.

In order to answer the question of whether SNORD50A/B is localized in endosomes we performed super-resolution microscopy to directly detect SNORD50A/B in cells. As previously described, we use pre-labeled fluorescent antibodies to KRAS and ARF6 to detect the recycling endosome and the localization of KRAS within it. To detect SNORD50A/B, we use anti-sense oligos that were previously validated in our first report to successfully knock down SNORD50A/B expression. To modify the ASOs for fluorescence microscopy, we generated the same sequences with 5' biotin modifications and used a streptavidin-fluorophore conjugate to generate a visible signal. As a control, we use anti-GFP targeting ASOs to define background signal. While it is certainly appropriate in other experiments, we did not use other snoRNA-targeting ASOs because the literature on the non-canonical functions of snoRNAs is actively expanding, and to our knowledge, no snoRNAs has been definitely excluded from localizing to the recycling endosome. We find as a result of these experiments direct visualization of SNORD50A/B in recycling endosome in detectable close proximity to KRAS (**New Supplemental Figure 8B**).

To answer the question of whether SNORD50A/B can be detected in the cytoplasm and endosomes, we had performed fractionation qPCR to quantify the relative quantity of each snoRNA by compartment. These experiments demonstrated detectable quantities of SNORD50A/B in both the cytoplasm and the membrane bound compartments, which is primarily endosomal (**Figure 4E,F**). SNORD50A appears to be more cytoplasmic while SNORD50B favors the membrane compartment, although both were within 2-fold expression by compartment. Notably in these experiments, the nucleus was separated from the cytoplasmic and membrane-bound compartments in the first purification step and excluded since there has not been consistently described function of KRAS within the nucleus, so this was unlikely to be relevant to our investigation.

While we appreciate the scientific question of concentrations of nucleolar/cytoplasmic/endosomal SNORD50A/B, we respectfully submit that the quantification of nuclear SNORD50A/B is beyond the scope of the current study. We did, however, ask the question of whether physiologically relevant levels of SNORD50A/B had been previously reported outside of the nucleus where snoRNAs perform their canonical function. Indeed, Huang et al. 2017 report that SNORD50A has non-canonical mRNA 3' processing inhibitory function in the cytoplasm (**PMID: 28911119**), further supporting extra-nuclear function of the snoRNA.

“Figure S8B, C show colocalization of proteins KRAS/SNAP23 and if I interpret the data correctly, the analysis is based on six and seven cells (right panel). This is way to indirect to support physiological binding of SNORD50. The same holds true for S4B.”

The Reviewer is correct that the data presented in previously **Figure S8B,C**, now **New Figure S8D,E** represents 3-4 measures per cell of 6 cells per condition. The purpose of the experiment was to explore and demonstrate a probable mechanism by which SNORD50A/B can displace KRAS-SNAP23, not to demonstrate physiologic binding between the snoRNA and protein – this is addressed elsewhere as is noted in clarified text. The technical limitations of both the acquisition and analysis of super-resolution microscopy also prohibit us from meaningfully expanding that dataset.

We do, however, support the physiologic direct binding in vivo of SNORD50A/B with KRAS in numerous other experiments. **Siprashvilli et al. 2016 Figure 2D** demonstrates in vivo cross-linking of KRAS to SNORD50A/B with great specificity as compared to other snoRNAs. UVC crosslinking as used in that experiment requires direct-angstrom interaction of the cross-linked amino acid of the protein and nucleotide of the RNA inside live cells (**PMID: 30068528**). Additionally, **Siprashvilli et al. 2016 Figure 2G** shows proximity ligation assays that result in fluorescently labeled rolling circle PCR amplifications of <40nm proximity between SNORD50A/B and KRAS. In our current work, we extend these findings by showing in **Figure 4E,F** that RAS-SNORD50A/B CLIP is highly specific to the membrane bound compartments. We also add new data in this revision **New Supplemental Figure 8B** (described in more detail above) demonstrating close spatial proximity of SNORD50A/B to KRAS in recycling endosomes.

Supplemental Figure 4B is similarly an adjunct to other experiments. The observation of KRAS localization changes is presented in **Figure 4A,B**, **Supplemental Figure S4A**, **Supplemental Figure 7A**, **New Supplemental Figure 8A,B,C,D,E**. Additionally, **Supplemental Figure 4B** is included as an observational control using GFP-tagged KRAS in live cells for those previous experiments to rule out the possibility that formaldehyde cross-linking interferes with antibody-mediated fluorescence microscopy.

“In response to 2.5, a specificity control, namely the expression of another SNORD is missing (SNORD expression construct flanked by two non-coding exons, a SNORD different from SNORD50)”

We thank the Reviewer for this excellent suggestion. In **Siprashvilli et al. 2016 Figure 2D** we had previously confirmed by live-cell KRAS CLIP that SNORD33 and SNORD83B do not physiologically interact with KRAS. We therefore proposed using these two non-coding RNAs as controls. To confirm that these RNAs were appropriate negative controls, we performed binding experiments of in-vitro transcribed SNORD33 and SNORD83B to protein microarrays of 9,125 recombinant proteins spotted in duplicate (**New Data Tables 8,9,10,11**). These experiments were performed under identical conditions as those previously reported for SNORD50A/B in **Siprashvilli et al. 2016 Supplemental Data**. As expected, SNORD33 and SNORD83B lacked significant in vitro binding capacity for KRAS.

SNORD33 and SNORD83B were then overexpressed in A549 in the same manner as previously described for SNORD50A/B (**New Supplemental Figure 8C**). Super-resolution microscopy was performed labeling KRAS, SNAP23, and ARF6 under these new conditions. We find that mean KRAS-SNAP23 distances in recycling endosomes is unchanged from mock infection to SNORD33 and SNORD83B overexpression, but significantly increased with SNORD50A/B overexpression (**New Supplemental Figure 8D,E**), thus suggesting that SNORD50A/B specifically displaces KRAS from SNAP23 in recycling endosomes.

Reviewer #3 (Remarks to the Author):

“The authors have addressed the raised concerns to my satisfaction”

We thank the Reviewer for the great suggestions in the prior rounds of review and collaborative effort in improving the manuscript.

REVIEWERS' COMMENTS:

Reviewer #2 (Remarks to the Author):

all concerns have been addressed